



# Carbon sequestration potential of street tree plantings in Helsinki

Minttu Havu[1], Liisa Kulmala[2,3], Pasi Kolari[1], Timo Vesala[1,3,4], Anu Riikonen[3], and Leena Järvi[1,5]

[1]Institute for Atmospheric and Earth System Research / Physics, University of Helsinki, Finland.
[2]Finnish Meteorological Institute, Helsinki, Finland.
[3]Institute for Atmospheric and Earth System Research / Forest Sciences, University of Helsinki, Finland.
[4]Yugra State University, 628012, Khanty-Mansiysk, Russia
[5]Helsinki Institute of Sustainability Science, University of Helsinki, Finland.

**Correspondence:** Minttu Havu (minttu.havu@helsinki.fi)

**Abstract.**

Cities have become increasingly interested in reducing their greenhouse gas emissions, and increasing carbon sequestration and storage in urban vegetation and soil as part of their climate mitigation actions. However, most of our knowledge on biogenic carbon cycle is based on data and models from forested ecosystems even though urban nature and microclimate are very different to those in natural or forested ecosystems. There is a need for modelling tools that can correctly consider temporal variations of urban carbon cycle and take the urban specific conditions into account. The main aims of this study are to examine the carbon sequestration potential of two commonly used street tree species (*Tilia x vulgaris* and *Alnus glutinosa*) and their soils by taking into account the complexity of urban conditions, and evaluate urban land surface model SUEWS and soil carbon model Yasso15 in simulating carbon sequestration of these street tree plantings at different temporal scales (diurnal, monthly and annual). SUEWS provides the urban microclimate, and photosynthesis and respiration of street trees whereas the soil carbon storage is estimated with Yasso. Both models were run for 2002–2016 and within this period the model performances were evaluated against transpiration estimated from sap flow, soil carbon content and soil moisture measurements from two street tree sites located in Helsinki, Finland.

The models were able to capture the variability in urban carbon cycle due to changes in environmental conditions and tree species. SUEWS simulated the stomatal control and transpiration well (RMSE<0.31 mm h$^{-1}$) and was able to produce correct soil moisture in the street soil (nRMSE<0.23). Yasso was able to simulate the strong decline in initial carbon content but later overestimated respiration and thus underestimated carbon stock slightly (MBE>-5.42 kg C m$^{-2}$). Over the study period, soil respiration dominated the carbon exchange over carbon sequestration, due to the high initial carbon loss from the soil after the street construction. However, the street tree plantings turned into a modest sink of carbon from the atmosphere on annual scale as the tree and soil respiration approximately balanced photosynthesis. The compensation point when street trees plantings turned from annual source to sink was reached faster by *Alnus* trees after 12 years, while by *Tilia* trees after 14 years. Overall, the results indicate the importance of soil in urban carbon sequestration estimations.



## 1  Introduction

The ongoing climate warming is caused by anthropogenic emissions of greenhouse gases (GHGs). A large proportion of these
emissions, especially carbon dioxide ($CO_2$), originate from urban areas (Marcotullio et al., 2013). In order to fight against
the climate crisis, significant amount of cities have declared themselves to be carbon neutral in the future decades. Carbon
neutrality in a city scale means that either GHG emissions and sinks are in balance or alternatively, part of the emissions are
compensated elsewhere. Urban green areas have been found to sequester up to 14 % (Vaccari et al., 2013; Hardiman et al.,
2017) of cities' GHG emissions. However, urban nature is highly diverse which brings a lot of uncertainty to the estimates.
In order for cities to reliably quantify their own carbon sinks to urban vegetation and soil, more information of the biogenic
carbon cycle in urban areas is required.

Urban trees can offer a variety of ecosystem services ranging from carbon sequestration to cooling of local temperatures,
stormwater mitigation, and improving air quality (Pataki et al., 2011; Pickett et al., 2011). The efficiency of the ecosystem
services depends on the local growing and climatic conditions of trees. In cities, trees are affected for example by urban heat
island effect (Oke, 1982), soil moisture availability, limited growth conditions and management practises (Dahlhausen et al.,
2018; Nielsen et al., 2007; Raciti et al., 2014). Quantifying the carbon storage and sequestration of urban trees has been
previously studied with field campaigns (Riikonen et al., 2017), biomass estimations (Stoffberg et al., 2010), remote sensing
(Myeong et al., 2006; Zhao and Sander, 2015), and most widely with GIS-based i-Tree software, including i-Tree Eco and
i-Tree Streets (Nowak and Crane, 2000). The i-Tree uses data on tree characteristics and estimates the carbon sequestration
and storage by biomass equations developed for urban trees based on US urban tree data. Most of the studies are from US
(McPherson et al., 2005, 2011), but studies outside of US have applied these models as well (Soares et al., 2011; Russo et al.,
2014). However, these methods are incapable of catching the correct response of urban biogenic carbon cycle to local conditions
and changes in climate, and thus lack high temporal resolution. In addition, the methods focus on urban trees, ignoring other
vegetation types and commonly urban soil altogether.

Urban land surface models (LSMs) can be used to simulate the carbon cycle in urban areas (e.g. SURFEX, Goret et al., 2019)
but commonly vegetation is treated in a separate tile without any interaction with the built surfaces. The photosynthesis, and
plant and soil respiration in interaction with urban surfaces were recently included to the urban land surface model SUEWS
(Surface Urban Energy and Water balance Scheme, Järvi et al., 2019) allowing to examine the net carbon sink of urban
vegetation. In SUEWS, photosynthesis is modelled with empirical canopy model that takes into account the local conditions
affecting the plant's stomatal control, such as, air temperature, specific humidity, soil moisture and shortwave radiation (Järvi
et al., 2019). Furthermore, plant and soil respiration can be modelled as an exponential dependence on temperature. The
urban land surface models focus on the exchange of carbon between vegetation and atmosphere, taking into account soil
respiration, on a local scale. Overall, LSMs are ideal for partitioning observed net $CO_2$ fluxes into anthropogenic and biogenic
components, particularly considering the effect of the interaction of urban structure and vegetation on the urban climate and
thus on carbon sequestration. LSM simulated carbon sinks can also be used to reduce uncertainties in satellite and atmospheric
in situ observation derived anthropogenic $CO_2$ emissions.



Urban soils can differ extremely from natural soils (Pickett et al., 2011) as they are usually man-made when the streets and parks are built. Also management practices, e.g., irrigation, litter removal and fertilization affect soil directly. Previous studies have shown that the soil organic carbon (SOC) stocks in urban soils vary widely (Lorenz and Lal, 2015), with most studies

showing urban soils containing more SOC than non-urban areas (Pataki et al., 2006; Pouyat et al., 2006; Raciti et al., 2012; Edmondson et al., 2012, 2014; Lindén et al., 2020), but contradicting results have also been published (Sarzhanov et al., 2017; Liu et al., 2016; Chen et al., 2013). The consensus has been that initially after construction, the soil loses carbon rapidly, but in the next few decades, the amount of SOC will increase more in urban soils than in the natural environment (Pataki et al., 2006). In addition to higher amounts of SOC, urban soil respiration has been found to be higher than in the natural environment (Kaye

et al., 2005; Pataki et al., 2006; Sarzhanov et al., 2015; Decina et al., 2016). Depending on the management practices, more or less litter i.e. carbon input can reach the soil. Turf grasses are usually irrigated, fertilized and clipped regularly through out the growing season leading to higher soil carbon (Pouyat et al., 2009). On the contrary, aboveground plant litter is usually taken away from gardens, parks and roadsides and therefore less aboveground carbon reaches the soil to decompose.

In addition to the size of the SOC pool, soil carbon decomposition depends also on temperature and precipitation (Davidson

and Janssens, 2006). Therefore, there exist multiple ecosystem soil decomposition models that are driven by climate, such as, Yasso15 (Viskari et al., 2020), CENTURY (Parton et al., 1988), Millennial (Abramoff et al., 2018) and ORCHIDEE-SOM (Camino-Serrano et al., 2018). Soil carbon models are developed especially for native ecosystems, such as forests, and for agricultural soils (Karhu et al., 2012). None, as of our knowledge, has been developed to simulate the complexity of urban soils and therefore, it remains still unclear whether these models are suitable for urban areas. So far, the CENTURY model

has been used to evaluate soil organic carbon for turf grass in golf courses (Bandaranayake et al., 2003) and simulate how clippings affect SOC storage (Qian et al., 2003). In addition, CENTURY simulations of lawn SOC were more successful when management practices were considered (Trammell et al., 2017). Recently, the Yasso model was used to estimate city wide SOC in Finland (HSY, 2021) but it lacked verification against measurements. Because the urban environment and management have a large impact on the soil carbon cycle, the use of these models in cities requires more testing.

The aim of this study is to use SUEWS and Yasso to estimate the carbon cycle dynamics in urban nature. We had two specific objectives: 1) to describe the diurnal, seasonal and interannual $CO_2$ flux dynamics by planted urban street trees, and 2) to describe the temporal dynamics of the organic carbon pool in their soil beneath. For the purpose, we evaluated the performance of both models using measurements from two street tree sites in Helsinki, Finland. On both sites, three different growing media were applied. The stomatal control model in SUEWS was parametrized to meet leaf-scale measurements of

street trees and verified against whole-tree transpiration of the trees whereas Yasso model was evaluated against SOC pools.

## 2 Materials and methods

SUEWS and Yasso models were used to simulate the two street tree sites in 2002–2016. The sites represent typical suburban neighborhoods of Helsinki.





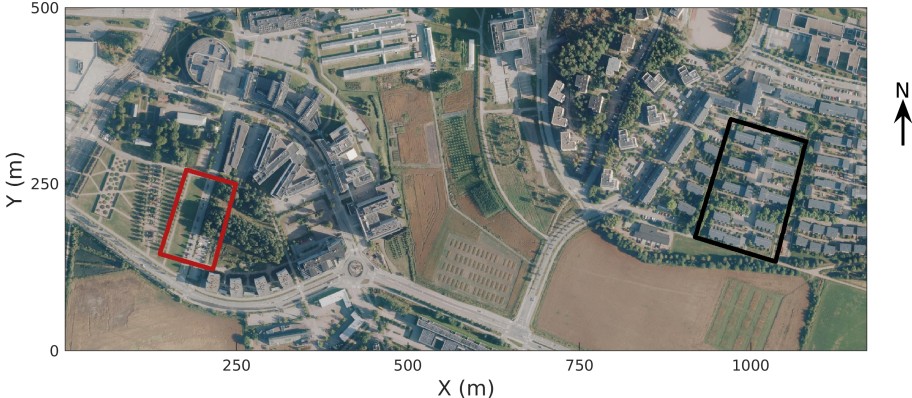

**Figure 1.** Study areas in Viikki, Helsinki (Kaupunkimittausosasto, Helsinki, 2019). The Tilia site is marked by a red square and the Alnus site by a black square.

## 2.1 Site description

In 2002, the city of Helsinki in collaboration with University of Helsinki established two street tree study sites in Viikki (N60°15', E25°03', Fig. 1, Table 1), 9 km northeast of the Helsinki city center, as part of the Viikki Street Tree Research project (2002–2016, Riikonen et al., 2011). The main aim of the project was to examine the impact of growing media on the growth and well-being of street trees, and during the study period, intensive monitoring of tree properties, gas exchange and soil carbon content was made. On one street (hereafter Tilia site), 15 *Tilia x vulgaris* Hayne trees were planted whereas on

another street (hereafter Alnus site), 22 *Alnus glutinosa* (L.) Gaertn. f. *pyramidalis* 'Sakari' trees were planted. Approximately, 15–30 $m^3$ and 45–50 $m^3$ rooting volumes were provided for each *Tilia* and *Alnus* tree, respectively. The spacing between the trees was 15 m for *Tilia* and 4–5 m for the *Alnus* trees. The Tilia site is surrounded by a park and office buildings, and the Alnus site by 2-floor apartment buildings. Hereafter, we call the unity formed by the trees and their growing media i.e. soil as street tree plantings. Tilia and Alnus sites can be characterized by Local Climate Zone (LCZ, Stewart and Oke, 2012) 9 and 6,

respectively.

Both sites had three different structural soils constructed as 1 m deep and 3 m wide layers. Soil 1 composition is mainly sand, clay and peat, soil 2 is composted of sewage sludge mixed with sand, pine bark and peat, and soil 3 is a mix of fine gravel, sand, clay, leaf compost and pine bark. Soils 1 and 2 are commercial soils, but soil 3 is a mixture done specifically for the research project. Riikonen et al. (2017) estimated initial loss-on-ignition (LOI) for each soil type. The initial LOI was 6,

20 and 4.4 % for soil 1, 2 and 3, respectively. The initial LOI, fine soil dry bulk density (BD) and stone matrix were measured in laboratory (Riikonen et al., 2011) and used in evaluating the SOC pools of the soils. On average, 32 % of the 1 m deep soil layer is fine soil and the averaged saturated soil water capacity of the fine soil is 45 %. The measured fine soil permanent wilting point (WP) is 6 %.



**Table 1.** *Site characteristics and model parameters for Tilia and Alnus sites in Viikki, Helsinki.*

| Variable | Tilia site | Alnus site |
|---|---|---|
| Latitude | 60° 13' 32.60' N | 60° 13' 35.58" N |
| Longitude | 25° 0' 46.34" E | 25° 1' 40.97" E |
| Time zone | 2 | 2 |
| Modelling height (m) | 31 | 31 |
| Altitude (m) | 5 | 5 |
| LCZ | 9 | 6 |
| $A$ (ha) | 1.50 | 2.19 |
| $fr_{build}$ | 0.02 | 0.20 |
| $fr_{paved}$ | 0.59 | 0.57 |
| $fr_{decid}$ | 0.23 | 0.21 |
| $fr_{bsoil}$ | 0.16 | 0.02 |
| $z_b$ (m) | 12.20 | 5.90 |
| $z_t$ (m) [a] | 5.48-8.46 | 7.14-16.66 |
| Trunk diameter at breast height (cm) [b] | 11.1-13.9 | 12.4-16.1 |
| Projected canopy area (m$^2$) [b] | 8.9-10.6 | 3.5-6.0 |
| Daytime population density (inh $\cdot$ ha$^{-1}$) [c] | 0.001 | 8.887 |
| Night-time population density (inh $\cdot$ ha$^{-1}$) | 0.001 | 109.590 |
| Traffic rate (veh km $\cdot$ m$^{-2} \cdot$ day$^{-1}$) [d] | 0.006 | 0.018 |

[a] Tree height grows exponentially through the years

[b] Measured in 2008–2011

[c] HSY (2011)

[d] HEL (2016)

## 2.2 Ecophysiological measurements

The portable gas exchange sensor (CIRAS-2, PP Systems, UK) was used to determine leaf-level responses of transpiration and $CO_2$ exchange to environmental drivers (light, $CO_2$). A total of 22–25 leaf samples located at different positions in the crown in six to seven trees of each studied species were measured during five field campaigns in 2007–2009 (Riikonen et al., 2011). The campaign measurements were normally carried out between 8 am and 4 pm. The measured light and $CO_2$ responses of leaf-level $CO_2$ exchange were scaled to stand level using the forest stand gas exchange model SPP (Mäkelä et al., 2006)

and meteorological measurements from Kumpula (See Sect. 2.3). The optimal stomatal control model (Hari et al., 1986) was used as the photosynthesis model in SPP. The stand-level photosynthetic responses were used to derive stomatal conductance parameters representative of *Tilia* and *Alnus* street trees in the SUEWS model (See Sect. 2.4.3).



To get estimation for whole-tree transpiration, sap flow $sf_m$ (l m$^{-2}$ h$^{-1}$ or mm h$^{-1}$) was measured with Granier type heat dissipation sensor pair (Hölttä et al., 2015) from three *Tilia* and three *Alnus* trees (Riikonen et al., 2016). The measured sap flow was divided with the projected canopy area (PCA) and averaged over the trees. The measurements were available for summers 2008–2011 and only months from June to August were used in this study to evaluate the SUEWS model. The time lag between the sap flow measurements, transpiration and environmental conditions varied between 30–90 min (Riikonen et al., 2016). The best fit between transpiration and sap flow measurements for the most cases was found with 60 min lag time and that was chosen for the whole study period. Comparing the water use of the different tree species, *Tilia* trees water use was approximately one fourth of the *Alnus* water use on a PCA basis.

Soil volumetric water content (SWC) also used to evaluate SUEWS model performance was measured at the depths of 10 and 30 cm below the surface with Theta-probes (ML2x, Delta T Devices Ltd., Cambridge, UK). SWC was averaged over different trees, soil types and depths separately for Tilia and Alnus sites. Overall, Tilia site had higher SWC as also the observed groundwater level was continuously high and the catchment area large, whereas Alnus site was fed mainly with local rainfall (Riikonen et al., 2011). For the summers from 2008 to 2011, the SWC was on average 27 and 13 % for Tilia and Alnus sites, respectively.

The soil carbon stock measurements used to evaluate Yasso model were available in 2002, 2005, 2008, 2011 and 2014 (Riikonen et al., 2017). The soil samples were collected in autumn from each soil type from depths varying between 30 to 90 cm. The soil carbon stock estimates were derived from the soil samples, where LOI was determined for each soil type. The proportion of carbon in the LOI was assumed 0.56 (Hoogsteen et al., 2015).

### 2.3 Meteorological measurements

Meteorological variables used to force the models with hourly resolution for years 2002–2016 were primarily from the nearby (4 km) SMEAR III urban measurement station in Kumpula (Järvi et al., 2009). Air temperature ($T_{air}$) (Pt-100, "in-house"), wind speed ($u, v, z$) (Thies Clima 2.1x, Gottingen, Germany) and incoming shortwave radiation ($K_\downarrow$) (CNR1, Kipp& Zonen, Delft, the Netherlands) were measured on top of a 31 meters high measurement mast. Air pressure (DPA500, Vaisala Oyj, Vantaa, Finland), relative humidity (HMP243, Vaisala Oyj), and precipitation (rain gauge, Pluvio2, Ott Messtechnik GmbH, Germany) were measured on the roof of a nearby building at 24 m above the ground. Additional precipitation measurements started in 2014 (PWD-11, Vaisala Oyj) and these were primarily used when available.

In order to create continuous meteorological forcing files for the modelled years, missing data from Kumpula were gap filled with observations from a station at Helsinki-Vantaa airport hosted by Finnish Meteorological Institute located 10 km northwest from Viikki. More detailed information of the gap filling procedure is given in Appendix A.

### 2.4 SUEWS

The Surface Urban Energy and Water balance Scheme (SUEWS) was originally developed to simulate the urban surface energy and water balance at a local or neighborhood scale (Järvi et al., 2011; Ward et al., 2016). The model includes several submodels for net all-wave radiation (Offerle et al., 2003), storage (Grimmond et al., 1991; Sun et al., 2017) and anthropogenic heat fluxes,





snow and irrigation (Järvi et al., 2014) to take urban features in the balances appropriately into account. Recently, the surface-atmosphere exchange of anthropogenic and biogenic $CO_2$ have been included to the model providing integrated information of the energy, water and $CO_2$ cycles in urban areas, including the impact of increased air temperatures on the water and $CO_2$ cycles (Järvi et al., 2019). This study used the most recent SUEWS version available V2020a. The model is forced with commonly
measured meteorological variables, such as, wind speed, wind direction, air temperature, pressure, precipitation and shortwave radiation. Specific site information are also needed in the model simulations, such as, surface cover fractions, and tree and building heights.

### 2.4.1   Biogenic $CO_2$ flux

The biogenic $CO_2$ flux components include the carbon uptake by photosynthesis ($F_{GPP}$) and carbon emissions by vegetation
respiration ($F_R$). Soil respiration can be included if integrated vegetation and soil parameters are used in the model runs. An empirical canopy-level photosynthesis model (Järvi et al., 2019) was used for the connection of transpiration to photosynthesis via stomatal conductance, and its dependency on local environmental conditions. $F_{GPP}$ ($\mu$mol m$^{-2}$ s$^{-1}$) for deciduous trees is calculated from

$$F_{GPP} = fr_{decid} F_{GPP,max,decid} LAI_{decid} f(T_{air}) f(\Delta q) f(\Delta \theta) f(K_{\downarrow}), \tag{1}$$

where the potential photosynthesis ($F_{GPP,max,decid}$) is scaled with leaf area index (LAI$_{decid}$, m$^2$ m$^{-2}$), surface cover fraction ($fr_{decid}$), and by the environmental response functions $f(T_{air})$, $f(\Delta q)$, $f(\Delta \theta)$, and $f(K_{\downarrow})$ on air temperature, specific humidity deficit, soil moisture deficit, and shortwave radiation, respectively. The functions have forms (Ward et al., 2016)

$$f(K_{\downarrow}) = \frac{K_{\downarrow}/(G_2 + K_{\downarrow})}{K_{\downarrow,max}/(G_2 + K_{\downarrow,max})}, \tag{2}$$

$$f(\Delta q) = G_3 + (1 - G_3)G_4^{\Delta q}, \tag{3}$$

$$f(T_{air}) = \frac{(T_{air} - T_L)(T_H - T_{air})^{T_C}}{(G_5 - T_L)(T_H - G_5)^{T_C}}, \tag{4}$$

where

$$T_C = \frac{(T_H - G_5)}{(G_5 - T_L)}, \tag{5}$$

and

$$f(\Delta \theta) = \frac{1 - \exp(G_6(\Delta \theta - \Delta \theta_{WP}))}{1 - \exp(-G_6 \Delta \theta_{WP})}. \tag{6}$$

Parameter $G_2 - G_6$ describe the responses of photosynthesis and stomatal conductance on each environmental variable. $K_{\downarrow,max}$ (W m$^{-2}$) is the maximum observed shortwave radiation, $T_L$ and $T_H$ (°C) are the lower and upper limits for temperature to





determine when photosynthesis and transpiration switch off, and $\Delta\theta_{WP}$ (mm) is the wilting point. The variables $\Delta q$ (g kg$^{-1}$), $K_\downarrow$ (W m$^{-2}$) and $T_{air}$ (°C) are given to the model as an input at the modelling height, typically well-above the urban surface, but SUEWS has an option to model local values of $\Delta q$ and $T_{air}$ at 2-m height (Sun and Grimmond, 2019) allowing to take into account the impact of local climate conditions on the spatial variability of $F_{GPP}$. $\Delta\theta$ (mm) is simulated within SUEWS (Järvi et al., 2017).

In SUEWS, $F_R$ increases exponentially with measured input or modelled local air temperature. Air temperature is used instead of soil temperature due to its common availability. $F_R$ ($\mu$mol m$^{-2}$ s$^{-1}$) is simulated with empirical constants $a$ and $b$ following

$$F_R = fr_{decid}max(a_{decid} \cdot exp(T_{air}b_{decid}), 0.6). \tag{7}$$

The lower limit of $F_R$ (0.6 $\mu$molm$^{-2}$s$^{-1}$) takes into account carbon emissions in winter that can not be achieved with the simple exponential model (Järvi et al., 2019). In this study, $F_R$ included only aboveground respiration as the soil respiration is determined with Yasso (see Sect. 2.5). In order to correctly simulate the carbon sequestration and respiration of street trees, the empirical parameters in both Eq. (1) and (7) are derived from urban leaf-level photosynthetic observations for deciduous street trees in Helsinki (Riikonen et al., 2011) (See Sect. 2.4.3).

### 2.4.2 Evapotranspiration

The latent heat flux ($Q_E$, W m$^{-2}$) including both evaporation and transpiration is calculated with the modified Penman–Monteith equation for urban areas (Grimmond and Oke, 1991)

$$Q_E = \frac{s(Q^* + Q_F - \Delta Q_s) + \rho c_p VPD/r_{av}}{s + \gamma(1 + r_s/r_{av})}, \tag{8}$$

where $Q^*$ (W m$^{-2}$) is the net all-wave radiation, $Q_F$ (W m$^{-2}$) the anthropogenic heat flux, $\Delta Q_S$ (W m$^{-2}$) the net storage heat flux, $\rho$ (kg m$^{-3}$) the density of air, $c_p$ (J kg$^{-1}$ K$^{-1}$) the specific heat capacity of air at constant pressure, VPD (Pa) the vapour pressure deficit, $s$ (Pa °C$^{-1}$) the slope of the saturation vapour pressure curve, $\gamma$ (Pa °C$^{-1}$) the psychrometric constant, $r_{av}$ (s m$^{-1}$) the aerodynamic resistance for water vapour and $r_s$ (s m$^{-1}$) the surface resistance. The surface resistance or its inverse surface conductance $g_s$ (m s$^{-1}$) depends on the same environmental factors as photosynthesis (Ward et al., 2016)

$$g_s = \frac{1}{r_s} = g_{max,decid}\frac{LAI_{decid}}{LAI_{max,decid}}fr_{decid}G_1 f(T_{air})f(\Delta q)f(\Delta\theta)f(K_\downarrow), \tag{9}$$

where the maximum conductance $g_{max,decid}$ is scaled with maximum leaf area index (LAI$_{max,decid}$), $fr_{decid}$ and the environmental response functions. $G_1$ (mm s$^{-1}$) is a constant obtained from latent heat ($Q_E$) and sensible heat ($Q_H$, W m$^{-2}$) observations and it connects stomatal conductance to canopy conductance.

### 2.4.3 Fitting environmental response functions

To get correct response of street trees to environmental factors in SUEWS, the environmental response functions ($f(T_{air})$, $f(\Delta q)$, $f(\Delta\theta)$, and $f(K_\downarrow)$) in Eqs. (1) and (9), were separately fitted for *Tilia* and *Alnus* trees using a non-linear least-square

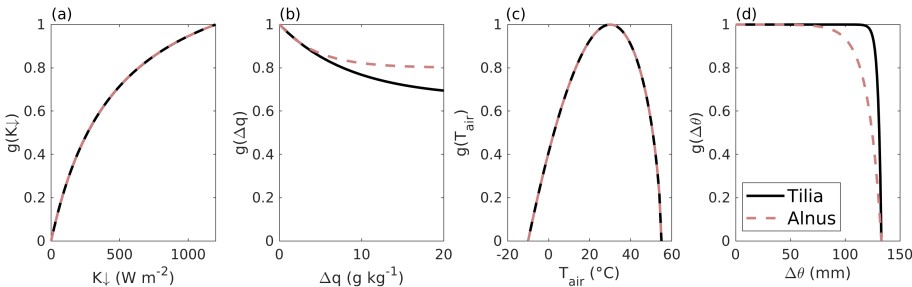

**Figure 2.** The fitted dependencies of surface conductance on environmental factors for (a) incoming shortwave radiation $K_\downarrow$, (b) specific humidity deficit $\Delta q$, (c) air temperature $T_{air}$ and (d) soil moisture deficit $\Delta\theta$ in SUEWS separately for Tilia (black solid line) and Alnus (red dashed line) trees.

method. In previous study at the Tilia site, similar fittings were made but only to fit $F_{GPP,max}$ and $f(\Delta q)$ assuming the other

function forms from park located in England (Järvi et al., 2019). To get more precise parameters to describe the behaviour of street trees, all the response functions were fitted against observations to get parameters $G_2 - G_6$ and $F_{GPP,max}$.

The previously calculated stand-level photosynthesis estimates for 2016 were used in the fitting as dependent variable while for independent variables observed $T_{air}$, $\Delta q$, and $K_\downarrow$ from Kumpula and SWC from the study sites were used. Fitting was made when $K_\downarrow > 10$ W m$^{-2}$ and $\Delta q > 1$ g kg$^{-1}$ as otherwise the stomatal conductance may deviate from the fits seen in

Fig. 2 (Bosveld and Bouten, 2001). This resulted all together 2492 data points. In the fitting, a bootstrapping method was used by randomly selecting 100 times 7/8th of the available observations with the final parameters calculated as medians with uncertainty from the fittings. Table 2 gives the fitted parameter values needed in Eqs. (2)–(6). In the calculation of $f(\Delta\theta)$ wilting point (WP) is needed to calculate the limit $\Delta\theta_{WP}$. A site specific estimate for $\Delta\theta_{WP}$ was calculated with soil information from Riikonen et al. (2011).

Figure 2 shows the environmental response functions and their dependence on the corresponding variable. The parameter values are $G_2 = 476.727 \pm 2.324$ W m$^{-2}$, $G_3 = 0.661 \pm 0.011$, $G_4 = 0.891 \pm 0.007$, $G_5 = 30.000 \pm 0.000$ °C, $G_6 = 0.361 \pm 0.042$ mm$^{-1}$, and $F_{GPP,max,decid} = 8.346 \pm 0.035$ $\mu$mol m$^{-2}$s$^{-1}$ for the Tilia site. Similarly for the Alnus site $G_2 = 474.483 \pm 2.046$ W m$^{-2}$, $G_3 = 0.800 \pm 0.004$, $G_4 = 0.901 \pm 0.010$, $G_5 = 30.000 \pm 0.000$ °C, $G_6 = 0.083 \pm 0.001$ mm$^{-1}$, and $F_{GPP,max,decid} = 13.178 \pm 0.073$ $\mu$mol m$^{-2}$s$^{-1}$.

The respiration parameters $a$ and $b$ in Eq. (7) are obtained by fitting canopy-level respiration estimates from the street trees for year 2016 against air temperature measurements from Kumpula. The estimations represent respiration from leaves and branches. To estimate whole tree respiration, one third of the canopy respiration was added to the values before the fittings to represent respiration from the trunk. Using bootstrapping method described above, for Tilia site, parameter values $a = 0.78 \pm 0.002$ and $b = 0.08 \pm 0.0001$, and for Alnus site $a = 1.11 \pm 0.003$ and $b = 0.08 \pm 0.0001$ are obtained.



**Table 2.** *SUEWS parameters used to simulate photosynthesis, respiration and transpiration of the studied street trees.*

| Parameter | Tilia site | Alnus site | Reference |
|---|---|---|---|
| $LAI_{decid,max}$ ($m^2\ m^{-2}$) | 4.80 | 4.80 | Breuer et al. (2003), Eschenbach and Kappen (1996) |
| Soil depth$_{decid}$ ($m$) | 1.00 | 1.00 | Riikonen et al. (2011) |
| Soil water storage capacity$_{decid}$ ($m$) | 0.14 | 0.14 | |
| $F_{pho,max,decid}$ ($\mu$mol m$^{-2}$s$^{-1}$) | 8.3463 | 13.1778 | This study |
| $g_{max,decid}$ (mm s$^{-1}$) | 3.1 | 8.7 | Breuer et al. (2003), Eschenbach and Kappen (1999) |
| $G_1$ | 3.5 | 3.5 | |
| $G_2$ | 476.7266 | 474.4833 | This study |
| $G_3$ | 0.6613 | 0.8001 | This study |
| $G_4$ | 0.8907 | 0.8013 | This study |
| $G_5$ | 30 | 30 | Ward et al. (2016), this study |
| $G_6$ | 0.3612 | 0.0827 | This study |
| $\Delta\theta_{WP}$ (mm) | 132 | 132 | This study |
| $K_{\downarrow,max}$ (W m$^{-2}$) | 1200 | 1200 | Järvi et al. (2014) |
| $T_L$ (°C) | -10 | -10 | Ward et al. (2016) |
| $T_H$ (°C) | 55 | 55 | Ward et al. (2016) |
| $a_{decid}$ | 0.78 | 1.11 | This study |
| $b_{decid}$ | 0.08 | 0.08 | This study |

### 2.4.4 SUEWS run

SUEWS was run around the street trees sites within modelling areas of 1.5 ha at the Tilia site and 2.19 ha at the Alnus site (Fig. 1). The first modelled year 2002 was used as a spin-up year leaving 2003–2016 for the analysis on carbon balance. Years 2008–2011 were used to evaluate the model against the street tree observations. The hourly meteorological forcing data were used to force the model, however the model calculations had a time step of 5 min. The surface cover fractions and building heights (Table 1) for both sites were obtained from an airborne laser scanning data with a resolution of 1 m (StromJan, 2020). The modelling areas had buildings, paved surfaces, bare soil, grass and deciduous trees. As SUEWS gives integrated evapotranspiration, photosynthesis and respiration for the whole simulation domain, grass surfaces present in the areas were set to impervious surfaces. This had a minor impact on modelled local air temperature (on average 0.16 °C warmer in summer) and humidity, and furthermore on tree functioning, but this is seen more suitable approach when model outputs are compared with tree observations.

The trees at both sites were planted in 2002 and as SUEWS does not currently include tree growth, information of the development of the trees during the modelled period are obtained from the local measurements. Tree height and maximum LAI are given to SUEWS as model input for each year whereas the seasonal development of LAI is based on growing degree





days within the model. The tree heights were measured from 2002 until 2011 (Riikonen et al., 2016) and as the tree growths

follows exponential curves, the same exponential growth was assumed for rest of the years. The maximum LAI for both *Tilia* and *Alnus* trees was set to 4.8 m$^2$ m$^{-2}$ as obtained for *Tilia cordata* in Breuer et al. (2003) and *Alnus glutinosa* in Eschenbach and Kappen (1996), respectively. The observations as such were not used for the maximum LAI as they present values for individual trees and not for neighborhood (stand) level as expected by SUEWS.

The vegetation type specific maximum stomatal conductance values ($g_{max,decid}$) needed in the model input are significantly

different between the two tree species. *Alnus glutinosa* have larger water use than *Tilia x vulgaris*. Similarly to maximum LAI values, $g_{max,decid} = 8.7$ mm s$^{-1}$ were chosen for Alnus site based on a study made in Germany (Eschenbach and Kappen, 1999), and $g_{max,decid} = 3.1$ mm s$^{-1}$ was chosen for Tilia site based on Breuer et al. (2003).

The modelled soil depth under the street trees was 1 m and soil water storage capacity 0.141 m was calculated from laboratory measurements. The amount of water in the top 1 m soil was not sufficient to maintain the high transpiration raters of *Alnus* trees.

This can be due to many different reasons, for example, that street trees may not receive enough drainage from paved areas in the model, or tree roots may reach deeper than 1 m, from where they may receive more water if they reach groundwater, which SUEWS can not take into account yet. In order to estimate tree transpiration correctly in Alnus site, a modified simulation with additional water input (0.06 mm h$^{-1}$) to represent the groundwater intake was made. The limit was chosen such that the soil does not dry and limit the modelled transpiration. The run without water input is hereafter called base run and the modified run

the final run (See Sect. 3.1.2).

## 2.5 Yasso

Yasso15 (Viskari et al., 2020) is the most recent version of the soil carbon decomposition model Yasso (Tuomi et al., 2009; Liski et al., 2005), where the rate of decomposition depends on climatic conditions and chemical composition of the soil organic matter. The model can be run as an annual or monthly basis. The annual precipitation, air temperature and air temperature

amplitude or monthly precipitation and monthly average air temperatures are needed as model drivers. The model simulates the change in carbon stock based on the balance between the decomposition of soil organic matter and possible litter input. The decomposition rate varies for the four carbon compound groups included in the model: compounds soluble in ethanol (E), or in water (W), and compounds hydrolysable in acid (A) and neither soluble nor hydrolysable at all (N). There is also a mass flow towards recalcitrant humus (H). Litter input can be added into the model, such as, leaf or fine root litter and woody

litter such as branches, stems and coarse roots. The AWENH ratios are defined for the initial soil carbon pool and for the litter input separately. The parameters for decomposition rates of different compounds are based on global litter decomposition measurements.

In this study, a monthly time-step was used to simulate the SOC at the study sites. The model was forced with 2-meter local air temperature estimations simulated by SUEWS and precipitation measurements from Kumpula, using the monthly

precipitation and mean temperature for each month. As the streets were build in 2002 and the initial soil carbon amount and composition were known, the initial carbon pool was given to the model. The decomposition rates for each chemical compound were estimated based on the soil composition (Table 3). The organic matter in soil 1 was peat, therefore AWENH fractions





**Table 3.** *AWENH fractions used in the Yasso model runs for the different soil types and for fine roots.*

|            | A      | W      | E      | N      | H      | Reference                                |
|------------|--------|--------|--------|--------|--------|------------------------------------------|
| Soil 1     | 0.0633 | 0.0077 | 0.0026 | 0.8421 | 0.0842 | Kalliokoski et al. (2019)                |
| Soil 2     | 0.618  | 0.049  | 0.023  | 0.311  | 0.000  | Heikkinen et al. (2021)                  |
| Soil 3     | 0.408  | 0.198  | 0.099  | 0.295  | 0.000  | Aleksi Lehtonen, personal communication  |
| Fine roots | 0.551  | 0.133  | 0.067  | 0.250  | 0.000  | Akujärvi et al. (2014)                   |

for peat were chosen (Kalliokoski et al., 2019). The decomposition matter in soil 2 was a mixture of peat, sewage sludge and pine bark but the shares of the components were not known. For the soil 2, we used AWENH values determined for a mixture of composted sludge (70 %) and peat litter (30 %) (Heikkinen et al., 2021). Finally, soil 3 had only leaf compost as a decomposition matter, therefore AWENH of birch leaves (Personal communication with Aleksi Lehtonen) were used.

The aboveground litter is assumed to contribute only a little in the urban SOC stock, because it is mostly removed. Therefore, we ignored it in this study. However, the litter input of fine roots needs to be taken into account as those naturally stay in the soil. The annual root litter input was estimated assuming that the fine root biomass equals that of leaves and the life time of fine roots was one year. The leaf biomass for the study trees was estimated in 2005, 2008 and 2011 (Riikonen et al., 2017). The missing years in between were linearly interpolated. The growth rate before the first and after the last observations were extrapolated using the growth rates estimated between the first two and last two measurement, respectively. The roots were assumed to be evenly spread in the soil volume which were approximately 20 m$^3$ and 48 m$^3$ for Alnus and Tilia, respectively. The annual estimates were assumed to evenly distribute over the months. The AWEN shares in the root litter were estimated to be as in Akujärvi et al. (2014) (Table 3) and carbon content in the fine root litter 50 %. The run without roots is hereafter called base run and the model run with roots the final run (See Sect. 3.2).

### 2.6 Model evaluation and statistics

The modelled soil moisture from SUEWS was evaluated against observations to examine the simulation of water balance in the model. Additionally, the performance of the surface conductance and photosynthesis models were evaluated against transpiration estimations from sap flow and leaf gas exchange measurements. The evaluation years were 2008–2011 when most of the measurements were available. Only months from June to August were included in the evaluation, however, in 2008 measurements were available only for July and August.

In order to compare the modelled and observed soil moisture, the modelled soil moisture deficits ($\Delta\theta$) were changed to soil water contents (SWC). The observed SWC is an average from depths 10 and 30 cm whereas the modelled SWC represents the average from the whole modelling area, excluding soil beneath buildings. The modelled soil depth depends on the surface type varying between 23 cm for paved areas and 1 m for the street trees. Thus, for the comparisons, both observed and modelled SWC have been normalized between 0 (dry soils) and 1 (wet soils) for each year.





In SUEWS the evapotranspiration for the whole simulation area is estimated from the modified Penman–Monteith model (Eq. 8). The sap flow measurements, against which SUEWS was evaluated, however provide estimation for transpiration of

street trees only. To overcome the different representativeness of the model and observations, comparisons between the two were only made for hours with no rain and over two hours after each rain event. The model output was scaled with street tree surface fraction to get the transpiration per tree area. Similarly, the observed sap flow was scaled with projected canopy area (PCA) to estimate the tree transpiration per tree area. The lag time between the sap flow measurements and the modelled transpiration was taken into account (See Sect. 2.2).

Simulated $CO_2$ uptake by photosynthesis and emissions by respiration were evaluated against leaf-level measurements that were scaled to canopy level for year 2016. These measurements were used for the stomatal conductance model parameter fittings in SUEWS and thus are not an independent data set. However, the comparison was made to show that SUEWS indeed reproduces similar responses to environmental conditions as the estimations from leaf-level measurements.

Yasso model simulations were compared with the carbon pool estimates driven from LOI based soil carbon contents. How-

ever, the first measurements point in 2002 is not used in the model evaluation as it was given to the model.

SUEWS can consider increase in tree height and increase of the canopy horizontally through surface cover fractions, but it cannot currently take densification of the canopy into account. This however must be taken into account when calculating the long-term carbon sequestration of street tree plantings. In the calculation of carbon sequestration of the street tree plantings for 2003–2016, the modelled tree gas exchanges were thus scaled with measured leaf area (LA) to obtain the densification of the

canopy. The canopy was allowed to grow (densify) between 2002 and 2008 after which the growth was assumed to cease due to regular pruning of the trees. The calculations for annual carbon sequestration and respiration were done based on how much space was allocated to one street tree. The soil respiration was scaled to 25 $m^2$ area typical for street trees and the trees were scaled to 9.5 and 4.7 $m^2$ for Tilia and Alnus site, respectively, based on estimations of canopy area from Riikonen et al. (2016). The soil respiration estimation was an average from the three soil types.

Common statistical metrics are used to evaluate the model performance, including, root-mean-square error (RMSE), normalized RMSE (nRMSE), mean bias error (MBE) and normalized MBE (nMBE). The RMSE is done with summed square of residuals:

$$\mathrm{RMSE} = \sqrt{\frac{\sum_{i=1}^{n}(y_i - \hat{y_i})^2}{n}}, \tag{10}$$

where $\hat{y_i}$ is modelled and $y_i$ measurement value. The normalization of RMSE is done with maximum and minimum values of

the observations:

$$\mathrm{nRMSE} = \frac{\mathrm{RMSE}}{y_{i,max} - y_{i,min}}. \tag{11}$$

The MBE is defined as follows:

$$\mathrm{MBE} = \frac{1}{n}\sum_{i=1}^{n}(\hat{y_i} - y_i) \tag{12}$$

and similarly to nRMSE, the nMBE is calculated using maximum and minimum values of the observations.





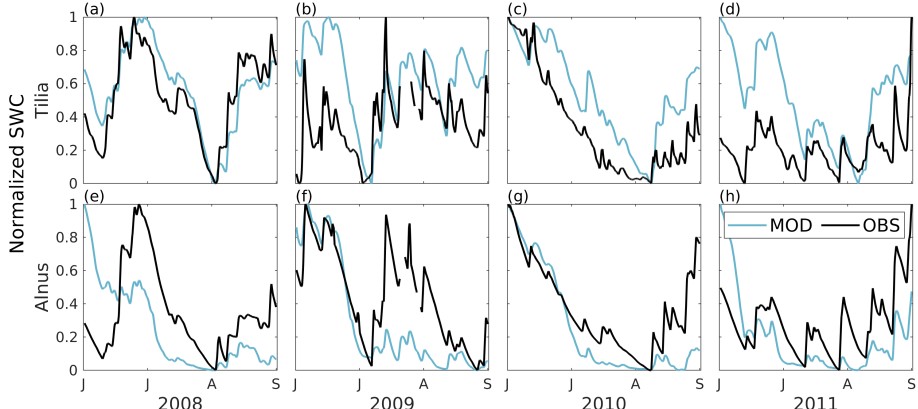

**Figure 3.** Modelled (MOD, blue) and observed (OBS, black) 1-day running mean of normalized soil water content (SWC) from June to August in 2008–2011. The normalization uses minimum and maximum values of modelled and observed SWC, respectively. The normalization is done separately for each year.

## 3 Results

### 3.1 SUEWS model performance

#### 3.1.1 Soil moisture

Simulated soil moisture covaried with the observations at both sites as shown in Fig. 3. The model performance was reasonably good, the nRMSE varied between 0.13–0.22 at the Tilia site, whereas at the Alnus site it varied between 0.16–0.23 (Table 4). In general, Tilia site was more moist than Alnus site. The model was not always able to catch the changes in SWC at Tilia site particularly in early summers 2009 and 2011 (Fig. 3b, d). At Alnus site, SUEWS was able to simulate SWC reasonably well (Fig. 3e–h). However, the base run showed on a few occasions exhaustion of the soil moisture under the street trees, which can be seen when the normalized modelled SWC approaches zero.

#### 3.1.2 Transpiration

SUEWS was able to simulate the observed diurnal dynamics of tree transpiration at Tilia site (Fig. 4 a). At the same time, SUEWS underestimated transpiration greatly at Alnus site when transpiration was compared with sap flow in the base run (Fig. 4 b). The model performance improved on a diurnal scale in the final run when a additional external water input of 0.06 mm h$^{-1}$ was included into the soil to represent the groundwater input to the tree roots.

Tilia site showed a slight morning maximum in the observations. The diurnal maximum of observed transpiration reached 0.27 mm h$^{-1}$. The model did not show the morning maximum and overestimated slightly the daytime transpiration with maxima values reaching 0.38 mm h$^{-1}$. At Alnus site, the modelled median transpiration reached 0.42 and 1.12 mm h$^{-1}$ for the



**Table 4.** *SUEWS model performance statistics for soil water content (SWC), transpiration and the $CO_2$ exchange components at the Tilia and Alnus sites.*

|  | Site | Year | RMSE | nRMSE | MBE | nMBE | N |
|---|---|---|---|---|---|---|---|
|  |  | 2008 | - | 0.13 | - | 0.04 | 2185 |
|  | Tilia | 2009 | - | 0.23 | - | 0.25 | 2012 |
|  |  | 2010 | - | 0.13 | - | 0.19 | 2185 |
| SWC |  | 2011 | - | 0.22 | - | 0.30 | 2185 |
|  |  | 2008 | 0.23 | 0.23 | -0.11 | -0.11 | 2185 |
|  | Alnus | 2009 | 0.21 | 0.21 | -0.14 | -0.14 | 2080 |
|  |  | 2010 | 0.16 | 0.16 | -0.11 | -0.11 | 2185 |
|  |  | 2011 | 0.20 | 0.20 | -0.10 | -0.10 | 2185 |
|  |  | 2008 | 0.10 | 0.34 | 0.06 | 0.21 | 820 |
|  | Tilia | 2009 | 0.12 | 0.27 | 0.02 | 0.05 | 1608 |
|  |  | 2010 | 0.13 | 0.14 | -0.05 | -0.05 | 1389 |
| Transpiration |  | 2011 | 0.11 | 0.34 | 0.08 | 0.25 | 1583 |
|  |  | 2008 | 0.23 | 0.17 | 0.07 | 0.05 | 1029 |
|  | Alnus | 2009 | 0.24 | 0.13 | -0.09 | -0.05 | 1691 |
|  |  | 2010 | 0.23 | 0.11 | -0.31 | -0.15 | 1380 |
|  |  | 2011 | 0.31 | 0.22 | 0.09 | 0.06 | 1585 |
| Respiration | Tilia | 2016 | 0.13 | 0.02 | 0.02 | 0.00 | 2147 |
|  | Alnus | 2016 | 0.18 | 0.03 | 0.08 | 0.01 | 2147 |
| Photosynthesis | Tilia | 2016 | 1.49 | 0.05 | -0.36 | -0.01 | 2147 |
|  | Alnus | 2016 | 2.32 | 0.05 | -1.16 | -0.03 | 2147 |

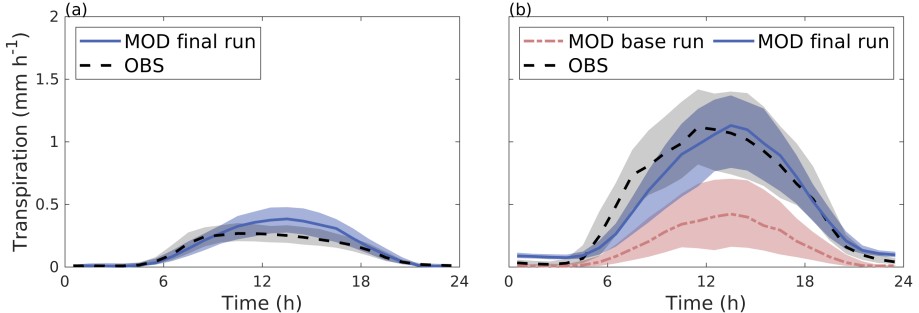

**Figure 4.** Median diurnal cycle of modelled transpiration (blue solid line), and transpiration estimated from observed sap flow (black dashed line) from June to August 2008–2011 for (a) Tilia site and (b) Alnus site. In panel b, the red line represent model simulation without an additional water source (the base run). The shadings are the 25/75th percentiles.





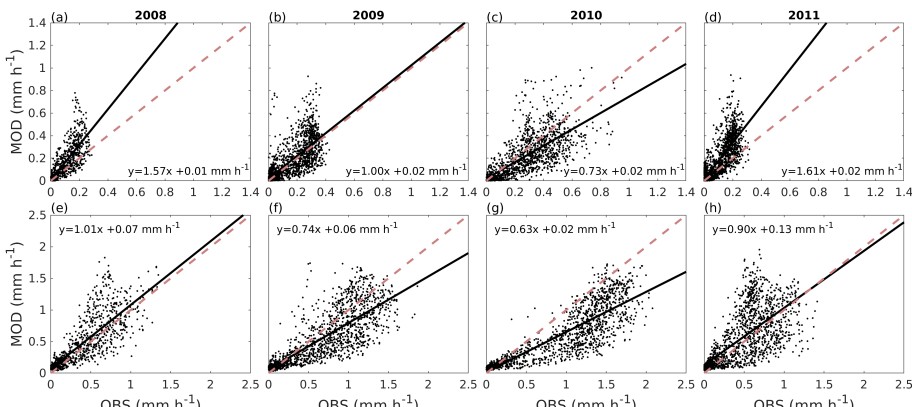

**Figure 5.** Correlation between hourly values of modelled transpiration (MOD) and transpiration estimated from sap flow measurements (OBS) from June to August for Tilia site (a–d) and Alnus site (e–h) for each year 2008–2011. The red dashed line is the 1:1 line and the black solid line represents the linear fit.

base run and final run, respectively, whereas the estimated transpiration from sap flow measurements was 1.12 mm h$^{-1}$ (Fig. 4b).

355    Figure 5 shows correlation between hourly values of modelled transpiration and transpiration estimated from sap flow measurements for summers from 2008 to 2011 separately for the two sites. The model performance varied between the different years. At Tilia site the nRMSE varied between 0.14–0.34 (in 2010 and 2011) whereas at Alnus site the performance was slightly better as the values ranged between 0.11–0.22 (in 2010 and 2011). Moreover, at the Tilia site the MBE varied between -0.05–0.08 mm h$^{-1}$ (in 2010 and 2011) whereas the Alnus site the performance was poorer as the values ranged between -0.31–0.09 mm h$^{-1}$ (in 2010 and 2011). Both sites showed higher transpiration in 2010 with measured 95th percentiles reaching 0.68 and 360    1.83 mm h$^{-1}$ for Tilia and Alnus site, respectively, whereas on other years the 95th percentiles remained below 1.48 mm h$^{-1}$. The modelled transpiration was slightly underestimating transpiration in 2010.

### 3.1.3  Photosynthesis and respiration

Figure 6 shows the median diurnal behaviour of photosynthesis and autotrophic respiration over June to August for 2016. Both the photosynthesis and respiration were larger for Alnus site. The daytime maxima photosynthesises were 22.5 and 35.9 365    $\mu$mol m$^{-2}$ s$^{-1}$ for Tilia and Alnus sites, respectively. Similarly, respiration maxima were 3.7 and 5.1 $\mu$mol m$^{-2}$ s$^{-1}$. The model performs well at both sites, nRMSE for respiration being 0.02 and 0.03 for Tilia and Alnus sites, respectively, and photosynthesis being 0.05 for both sites. At Tilia site the MBE of respiration was 0.02 $\mu$mol m$^{-2}$ s$^{-1}$ whereas at Alnus site it was 0.08 $\mu$mol m$^{-2}$ s$^{-1}$. Furthermore, the MBE of photosynthesis were -0.36 and -1.16 $\mu$mol m$^{-2}$ s$^{-1}$ for the Tilia and Alnus sites, respectively.





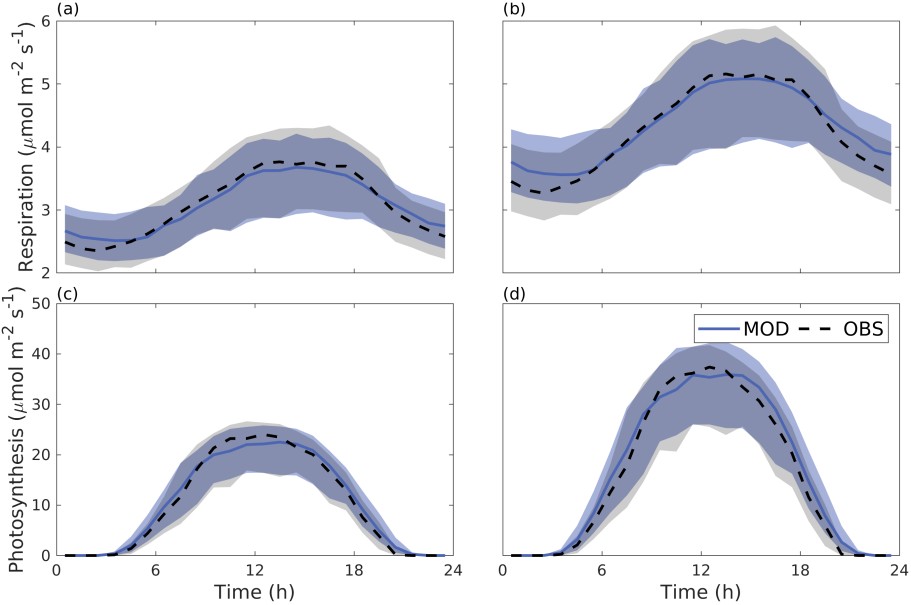

**Figure 6.** Median diurnal cycle of modelled (blue line) and observed (black dashed line) $CO_2$ emissions in tree respiration (a–b) and photosynthesis (c–d) from June to August 2016 for Tilia site (a,c) and Alnus site (b,d). The shadings show the 25/75 percentiles.

### 3.2 Yasso model performance

Overall from 2002 until 2016, the soil carbon pool decreased from 14.5, 27.9 and 9.6 kg C $m^{-2}$ to 5.1, 4.5 and 1.7 kg C $m^{-2}$ for the Tilia site for soil 1, 2 and 3, respectively, and to 5.7, 5.4 and 2.2 kg C $m^{-3}$ for the Alnus site (Fig. 7). Yasso model performance was the highest in soil 3 (Table 5). Yasso underestimated the soil carbon pool in soil 2 at both sites whereas the performance was mixed in soil 1 (Fig. 7). In general, the nRMSE ranged from 0.59 to 0.88 at the Tilia site, indicating better model performance than at the Alnus site, with values ranging from 0.73 to 1.36 (Table 5). Overall, the nMBE showed also better performance at the Tilia site, with values ranging from -0.91 to -0.75, whereas at the Alnus site, the range was from -1.63 to 2.21. The role of decomposing fine roots was small and barely detectable before the later phase of the simulation period as seen in the model run with roots deviating very little from the base run without roots (Fig. 7).

### 3.3 Carbon sequestration

The seasonal distribution of tree gas exchange and soil respiration slightly varied between the years (Fig. 8). The tree canopy area grew until 2008, after which the canopy was regularly pruned, and the annual changes in carbon sequestration and tree respiration were then mainly due to the prevailing weather. Autotrophic respiration was at its highest in July, while photosynthesis peaked in either June or July depending on the year. In 2010, the model estimated the highest monthly autotrophic respiration rates in July, with values 0.16 and 0.22 kg C $m^{-2}$ $month^{-1}$ for Tilia and Alnus sites, respectively. However, the maxima photosynthesis values were simulated in July 2014, with values 0.39 and 0.63 kg C $m^{-2}$ $month^{-1}$ for Tilia and Alnus




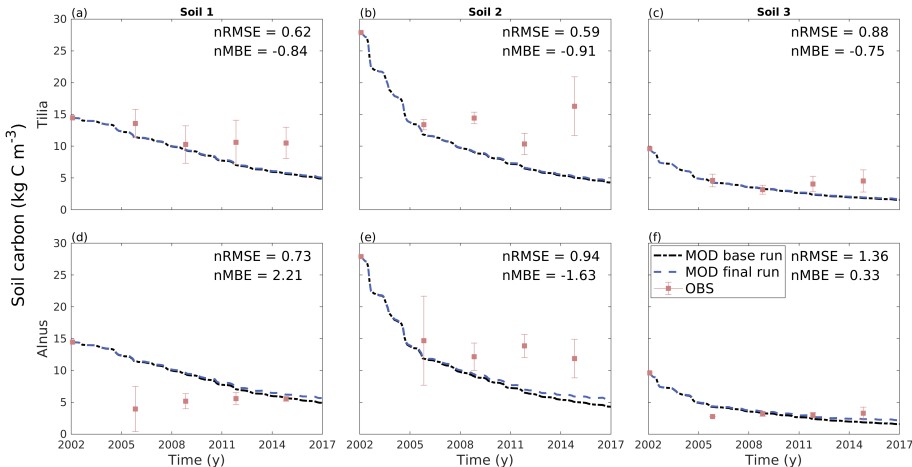

**Figure 7.** Modelled monthly soil carbon stock using Yasso without roots (dashed black line) and with roots (dashed blue line) from 2002 to 2016, and measured average LOI based soil carbon stock estimations (± SD) (red dots) for the three studied soil types at the Tilia site (a–c) and for the Alnus site (d–f).

**Table 5.** *Yasso-model performance statistics for soil carbon stock at the Tilia and Alnus sites by soil type.*

| Site | Soil | RMSE | nRMSE | MBE | nMBE |
|------|------|------|-------|------|------|
| | Soil 1 | 2.05 | 0.62 | -2.78 | -0.84 |
| Tilia | Soil 2 | 3.53 | 0.59 | -5.42 | -0.91 |
| | Soil 3 | 1.27 | 0.88 | -1.08 | -0.75 |
| | Soil 1 | 1.20 | 0.73 | 3.62 | 2.21 |
| Alnus | Soil 2 | 2.63 | 0.94 | -4.56 | -1.63 |
| | Soil 3 | 0.74 | 1.36 | 0.18 | 0.33 |

sites, respectively. Leaf onset begun at different times in different years depending on the simulated growing degree days, leading to a difference of up to 20 days in the model simulations. This is most evident in May 2015, when photosynthesis was 0.16 kg C m$^{-2}$ month$^{-1}$, which is only 55 % of the largest photosynthesis in May (in 2016). However, photosynthesis did not differ from other years on an annual basis because the growing season lasted longer in 2015, with vegetation remaining more active even in August when compared to other years. Soil respiration estimations (Fig. 8e, f) were higher in the initial years after the street construction. In July 2004, the model estimated highest soil respiration rates of 0.73 kg C m$^{-2}$ month$^{-1}$. After the initial soil carbon loss, the maximum monthly values ranged between 0.08 and 0.26 kg C m$^{-2}$ month$^{-1}$. According to the model, the highest monthly values could be reached from May to October, depending on the year. The variability in the seasonality of soil respiration is due to both temperature and moisture. In June 2010, the average monthly temperature was

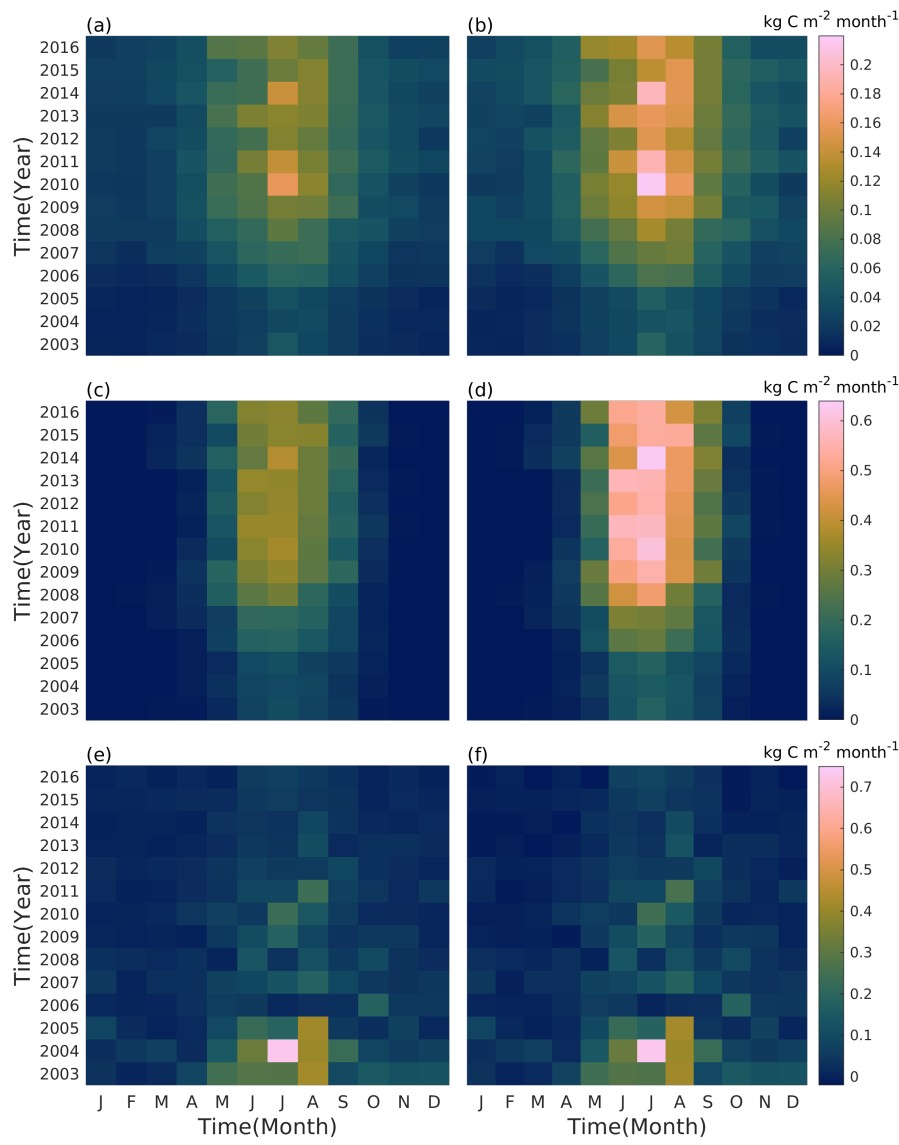

**Figure 8.** Estimated monthly street trees respiration (a,b), photosynthesis (c,d) and soil respiration (e,f) at the Tilia (a,c) and Alnus (b,d) sites during the simulation period (2003–2016).

exceptionally high 22.5 °C, however in August 2011, the monthly precipitation amount was high at 253.5 mm, leading to high soil respiration in both cases.

Over the whole study period (2003–2016), uptake by photosynthesis of the trees increased while the emissions from soil decreased (Fig. 9). As a result, the sites turned from annual $CO_2$ sources to climate neutral or even small sinks. The estimated annual uptake by photosynthesis varied between the years from 3.55 to 13.44 kg C year$^{-1}$ per tree for Tilia site and from 2.68





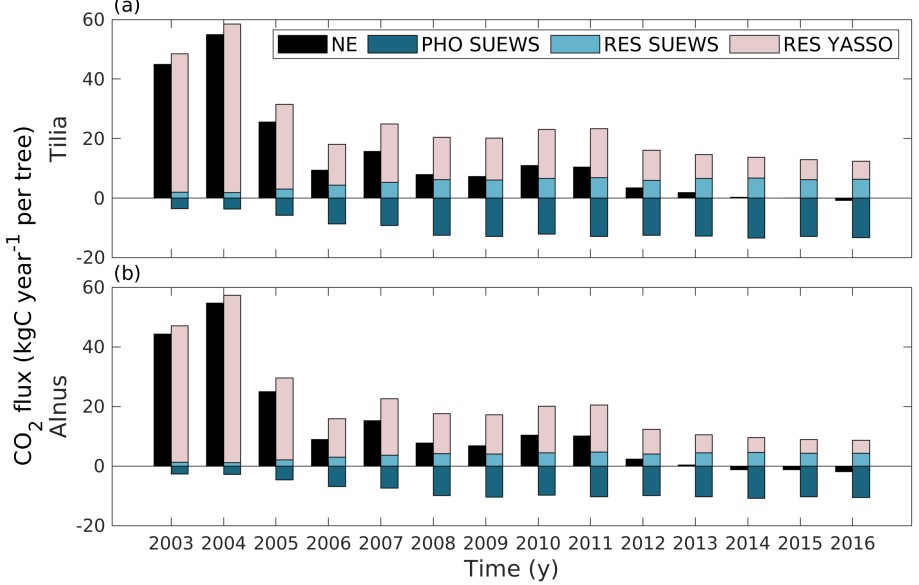

**Figure 9.** Estimated annual net exchange (NE, black) of street tree plantings, $CO_2$ uptake by photosynthesis (PHO SUEWS, dark blue) and emissions from tree respiration (RES SUEWS, light blue) simulated with SUEWS, and emissions from soil respiration simulated with Yasso (RES Yasso, light rose) at the Tilia (a) and Alnus (b) sites. Here, positive values indicate release of $CO_2$ to the atmosphere and negative values uptake from the atmosphere.

to 10.73 kg C year$^{-1}$ per tree for Alnus site. Similarly, respiration from trees varied between 1.87 and 6.80 kg C year$^{-1}$ per tree for Tilia site and 1.22 and 4.68 kg C year$^{-1}$ per tree for Alnus site. Soil respiration varied from 6.16 to 56.68 kg C year$^{-1}$ per tree for Tilia site and from 4.41 to 56.21 kg C year$^{-1}$ per tree for Alnus site. Overall, the net exchange (NE) of street tree plantings varied between -0.86 and 54.92 kg C year$^{-1}$ per tree for Tilia site and -1.82 and 54.70 kg C year$^{-1}$ per tree for Alnus site.

## 4  Discussion

In this work, we estimated the $CO_2$ exchange dynamics in common urban street trees and their growing media using validated models. We found that these ecosystems turned from source to sink of atmospheric carbon on annual level during the first 14 years after soil preparation and plantation of the trees. Cumulatively over the years, these street tree plantings would not become sinks until 30 years after the streets were built (Riikonen et al., 2017). Commonly used methods to asses the carbon sequestration of street trees, such as i-Tree, estimate the sink strength with biomass equations and growth rate estimations (Nowak and Crane, 2000). However, these methods are unable to provide high temporal variations. Furthermore, these studies have been focusing mostly on the carbon cycle of trees, leaving the soil carbon out of the estimations. The models used in this





study allow to consider temporal variations in urban carbon sequestration and respiration by vegetation and soil, and can take climate and local meteorological conditions into account in their estimations.

The carbon sequestration by the tree and soil beneath ranged from a strong carbon source to the atmosphere in the initial years (54.9 kg C year$^{-1}$ per tree) to a weak carbon sink at the end of the simulation period (-1.8 kg C year$^{-1}$ per tree). In the initial years after construction, the high soil carbon decomposition dominated the gas exchange. At the latest stages of the study period, the soil respiration roughly equalled tree respiration (approximately 5 kg C year$^{-1}$ per tree), and photosynthesis balanced these two components. These results highlight the importance of soil and its respiration in urban carbon balance,

which is often neglected in urban studies but as shown can have similar magnitude as tree carbon sequestration.

### 4.1   Dynamics of tree carbon gas exchange

We found that tree $CO_2$ exchange varied between days, seasons and years due to changes in environmental factors, tree species and tree size. The diurnal cycle of photosynthesis was mainly driven by the changes in incoming shortwave radiation, limiting the uptake at night time and on cloudy days. Additionally, the decrease in air humidity slightly limited the uptake at day time.

The seasonal variability was driven by variations in incoming shortwave radiation, air temperature and LAI whereas the year-to-year variability was driven by changes in air temperature and LAI only. In this study, the size of the tree canopy was assumed to remain constant after 2008, which is why the annual variations in carbon sequestration and tree respiration thereafter were mainly determined by prevailing weather. These street trees had access to water outside the growing medium, and therefore, the top 1 meter soil moisture did not limit $CO_2$ uptake by photosynthesis in this study.

Here, the annual tree respiration varied between 1.2 and 6.8 kg C year$^{-1}$ per tree and photosynthesis ranged between 2.7 and 13.4 kg C year$^{-1}$ per tree. In the last simulation year 2016, the net uptakes were 7.0 and 6.2 kg C year$^{-1}$ per tree for Tilia and Alnus sites, respectively. These estimations are lower than those resulted by other methods used to estimate carbon sequestrated by street trees in Europe. Russo et al. (2014) used models (UFORE and CUFR Tree Carbon Calculator), allometric equations and field data to estimate the average aboveground carbon sequestration of street trees in Bolzano, Italy, ranging from 12.1 to

17.4 kg C year$^{-1}$ per tree. Moreover, in Lisbon, Portugal, the street trees were estimated to sequester 43.1 kg C year$^{-1}$ per tree (Soares et al., 2011). However, the street trees grew in warmer temperate zone and were probably more mature and therefore, could sequester more carbon than the younger trees examined in this study.

    Tree biomass equations have been used to estimate the carbon accumulated to woody biomass, roots and leaves in 2003–2011 for the same street trees as used in this study. Riikonen et al. (2017) estimated that 26.1 and 38.2 kg C per tree for *Tilia* and

*Alnus* trees, respectively, was sequestrated during the first 10 years after planting. Correspondingly, 39.4 and 35.9 kg C per tree was estimated to accumulate based on the balance between simulated tree respiration and photosynthesis during the decade. However, respiration from roots was not taken into account in these simulations, which would decrease the accumulated carbon estimations. Moreover, urban biomass estimations have still uncertainty and Riikonen et al. (2017) noted that the estimation for *Tilia* trees might be an underestimation. Furthermore, i-Tree model has been used to estimate the carbon sequestration of

potential *Tilia* trees in Helsinki, using weather from Maine, USA (Ariluoma et al., 2021). The sequestration potential in 50 years was at best 1.7 t $CO_2$, corresponding on average to 7.6 kg C year$^{-1}$ per tree. The estimation was possibly overestimating




the carbon sequestration potential in Helsinki as the weather from Maine had more precipitation than what has been observed in Helsinki. In addition, how the models handle leaves varies depending on the method, as in these streets, we assume that all the leaves end up out of the simulation area, so their decomposition is not taken into account. Overall, the annual carbon
sequestration estimated with i-Tree was close to the estimations for *Tilia* trees in this study.

## 4.2    SUEWS performance and tree measurements

We found that SUEWS is able to simulate evapotranspiration dynamics correctly even though the study sites greatly differed in soil water availability. It is reported that *Alnus glutinosa* trees tend to have deep roots that can access groundwater (Claessens et al., 2010) and therefore, trees are not dependent only on precipitation but they can access to deep water sources. Our study
supports the phenomenon as the modelled transpiration at the Alnus site notably improved when an external water input was fed into the soil at the same time when without additional water, soil moisture in top layer was simulated well. Therefore, the possible existence of unidentified water pools might complicate further simulations of urban photosynthesis in soils with access to groundwater.

     Modeling photosynthesis is a relatively new addition to the SUEWS model (Järvi et al., 2019), combining evapotranspiration
and photosynthesis with stomatal opening. Model parameters $G_1 - G_6$ have been previously fitted against surface conductance values estimated from observed latent and sensible heat fluxes (Järvi et al., 2011; Ward et al., 2016), representing integrated conductance for all surface types. The effect of evaporation is eliminated by doing the parameter fittings only for dry conditions. These kind of general parameters represent the environmental response functions for all vegetation types compared to the method used in this study, where the parameters represent only street trees. Compared to general parameters derived from eddy
covariance measurements from Swindon, England (Ward et al., 2016) ($G_2 = 200$ W m$^{-2}$, $G_3 = 0.13$, $G_4 = 0.7$, $G_5 = 30$ °C, $G_6 = 0.05$ mm$^{-1}$, $\Delta\theta_{WP} = 120$ mm), $f(\Delta q)$ parameters $G_3$ and $G_4$ show significant difference. $\Delta q$ seems to be less relevant for street trees, however extreme dry conditions were not reached in the fitting period, which could affect the fitted parameters. The same behaviour was found in Riikonen et al. (2016), where they studied the $\Delta q$ relation to sap flow measurements. $f(K_\downarrow)$ is slightly more restricting for street trees than the general parameters. $f(T_{air})$ is the same for the general parameters as for
the street trees, because the shape and upper and lower limits are the same. The peak air temperature $G_5$ does not change as this high temperatures are rarely measured in Helsinki. $\Delta\theta_{WP}$ is slightly smaller for the Swindon site than what was estimated here. $f(\Delta\theta)$ for the general parameters is similar to the Alnus site.

     The dependencies of the different trees on $K_\downarrow$ and $T_{air}$ are very similar whereas clearly different responses on $\Delta q$ and $\Delta\theta$ are seen. $\Delta q$ relation to stomatal conductance has already been reported to be smaller for these street trees especially for the
Alnus site (Riikonen et al., 2016), whereas, in both sites, the soil moisture is expected to have little effect until significant deficit is reached. Especially on Tilia site, SWC is high and therefore no clear dependence to $\Delta\theta$ is found. The high soil water availability can also affect the stomatal conductance response to $\Delta q$, as even in dry air conditions, the trees have access to water in soil.

     Carbon sequestration and evapotranspiration both depend on the tree leaf stomata control. In this study, leaf-level gas ex-
change measurements were used to parameterize the stomatal control model in SUEWS, whereas sap flow measurements were





used to evaluate the functionality of the model. However, both measuring methods have known uncertainties. The leaf-level photosynthetic responses were not used as such, but were scaled to canopy-level with a forest stand gas exchange model SPP (Mäkelä et al., 2006). The measurements were made manually, so no continuous measurement data were available, but rather continuous photosynthesis data were created separately with SPP. For further research, automatic chambers would be rec-

ommended to get more realistic environmental response functions. The Granier type heat dissipation method (Granier, 1987; Hölttä et al., 2015) used in this study to measure sap flow and estimate whole-tree transpiration has some uncertainties, caused by method related issues, such as, the sensors respond slowly to the changes in flow rate, and by tree related issues, such as, the water stores in trees itself are utilized (Clearwater et al., 1999; Burgess and Dawson, 2008). These issues in the measurement method lead to time lag between the measured sap flow and the actual tree transpiration, and moreover, with meteorological

conditions affecting the transpiration. Riikonen et al. (2016) estimated the time lag for the street trees to range between 30 and 90 min depending on the year. Here, the average of 60 min was used for all cases, which may lead to a slight error. The *Tilia* trees showed a slight morning maximum in the observations, which might be due to transpiration from internal water reservoirs in the tree trunk. Furthermore, the observed sap flows may not be accurate representation of tree transpiration as the sensor location may not represent the whole tree trunk. However, Riikonen et al. (2016) estimated the possible overestimation to be

21 % at the highest. The sap flow values also varied between the measurements years, partly due to meteorological conditions. In 2010, the sap flow values could be at times twice as high than other years, due to higher air temperature and increased VPD observed that year. However, long-term measurements have some uncertainty, as trees grow, the sensors may be buried more deeply, leading to changes in the flow rates (Moore et al., 2010).

### 4.3  Soil carbon

Here, we demonstrated the relative importance of soil carbon in the carbon cycle of street trees. Cities have already used soil carbon models to estimate their soil carbon stocks, but relatively few studies exist about the applicability of these models to urban soils (Bandaranayake et al., 2003; Qian et al., 2003; Trammell et al., 2017). We showed that Yasso soil model is mainly able to simulate the initial decrease in soil carbon pool after planting of trees but there seems to be increasing misfit over the simulation period. The reasons behind remain unsolved in this study but we assume that the differences arises from unknown

initial AWENH of the soil substrates, spatially limited sampling of soil carbon pool and possibly overestimated soil moisture on paved systems. Next, we discuss these in detail.

Yasso simulates the decomposition of soil carbon depending on the solubility of the carbon compounds. The used AWENH fractions were based on qualitative description of the soil composed of different organic materials (Riikonen et al., 2017). Their proportions in the mixture, such as the share of peat, were unclear and therefore leading to uncertainty in the initial AWENH.

Further, setting of these initial fractions had high impact to the model results. For example, bark was ignored in the soil 2 as we assumed the share of it to be minor but the lack of it in the model runs might explain some of the underestimation in comparison with the measurements. On the other hand, the soil measurements have also a large uncertainty, as they were spatially measured only from two locations even though vertically from multiple depths. The samples were taken app. 2–3 meters from trees whereas we simulated the whole soil volume where the distance especially between the *Tilia* trees were





notably higher. According to the measurements, the soil carbon pool was stable or even increasing 7–15 years after the planting. Such finding in nature can result only from notable litter input, notable decrease in decomposition of organic matter or most likely combination on those two.

In the simulations, the fine roots had a minor impact in the soil carbon stock as the study trees were still young and thus the root biomass low. As the fine roots were assumed to be evenly spread in the model runs, the simulated fine root litter
input and decomposition represent an average from the whole soil volume. In nature, fine roots probably are denser close to the trees i.e. at the sampling locations than further away. Besides, high root mass decreases soil moisture and therefore also the decomposition rate. Higher root litter input and decreased decomposition rate at the sampling locations could cause the observed underestimation in the model simulation in the long run. With current knowledge, quantifying the fine root litter input is difficult as the amount and the turnover rate are still unknown especially in urban areas. The turnover rates have been
estimated to vary between one to nine years in forest ecosystems (Matamala et al., 2003) and therefore, future estimations would benefit from studies revealing more accurate root lifetime in urban ecosystems.

The forcing meteorology for Yasso was generated from the 2-m local air temperature simulated by SUEWS to get the local temperatures. Local temperatures vary spatially in urban areas, because build environments tend to warm more, and vegetative environments cool down because of evapotranspiration (Oke, 1982). However, the study sites in Viikki are similar to the
measurement site in Kumpula, so the difference between measured air temperature from Kumpula and the modelled local temperatures remained small. In theory, increased soil temperature would lead into increased decomposition of soil organic matter. At the same time, the role of soil moisture is more complex as the decomposition is decreased both in high and low soil moisture conditions (Moyano et al., 2012). Yasso soil carbon model is driven by precipitation but in these kind of paved systems, the soil moisture might be lower than expected as notable part of the water never enters the soil volume. Changing the
drivers belowground would probably lead to improved model performance but on the other hand, observations of soil moisture and temperature are rare. Nevertheless, further efforts are needed in studying the role of soil moisture in the decomposition of urban soil carbon pool.

The estimated SOC densities in 2016 ranged from 1.7 to 5.7 kg C m$^{-2}$, mostly depending on the soil type. Soils 1 and 2 reached similar SOC in 2016 (4.5–5.7 kg C m$^{-2}$) even though the initial SOC was almost twice as high for soil 2. These street
soil estimates are much lower than those previously measured in the parks of city of Helsinki (10.4 kg C m$^{-2}$; Lindén et al. (2020)) and even lower than forest soils in Finland (6.3 kg C m$^{-2}$; Liski et al. (2006)). However, direct comparison between SOC estimations can be challenging due to different soil types, vegetation and age. On the other hand, limited amount of new carbon enters the soil of these streets, which may explain part of the difference. The time of construction or renovation of the park had a major impact on SOC (Scharenbroch et al., 2005; Setälä et al., 2016), as also Lindén et al. (2020) found in the parks
of the city of Helsinki where SOC accumulation stabilized after 50 years. The effect of construction of the streets is clearly seen also from the street SOC estimations. The estimations show a decrease of SOC during the study period as the root litter input is not enough to stabilize the decomposition of SOC. Compared to other urban soil studies outside of Finland, the average SOC storage in greenspace was 9.9 kg C m$^{-2}$ in Leicester, UK (Edmondson et al., 2014), which show similar estimates as parks in Helsinki. However, in warmer climates, the estimated SOC values have been lower. In Singapore, under turfgrass, the





SOC was estimated to be 2.0 kg C m$^{-2}$ (Velasco et al., 2021). Furthermore, in Auckland, New Zealand, parkland soils were estimated to have 4.8 kg C m$^{-2}$ and urban forest soils 2.7 kg C m$^{-2}$ (Weissert et al., 2016).

The maximum monthly soil respiration estimates varied between 0.08 and 0.26 kg C m$^{-2}$ month$^{-1}$ after the high initial carbon loss, which corresponds to 2.5 and 8.1 $\mu$mol CO$_2$ m$^{-2}$ s$^{-1}$. These estimates compare reasonably well to previous research on soil respiration in urban areas. In greater Boston's residential areas (Decina et al., 2016), the soil respiration

of urban forests, lawns, and landscaped cover types were 2.6, 4.5, and 6.7 $\mu$mol CO$_2$ m$^{-2}$ s$^{-1}$, respectively. In Singapore, turfgrass soil respiration was measured to be an average 2.4 $\mu$mol CO$_2$ m$^{-2}$ s$^{-1}$ and highest mean value of 4.4 $\mu$mol CO$_2$ m$^{-2}$ s$^{-1}$ (Velasco et al., 2021). No seasonal trends were observed as the tropical weather is favourable to constant soil respiration. In New Zealand, median soil respiration was for parklands 5.2 $\mu$mol CO$_2$ m$^{-2}$ s$^{-1}$ and for urban forest sites 4.5 $\mu$mol CO$_2$ m$^{-2}$ s$^{-1}$ (Weissert et al., 2016).

## 5   Conclusions

Quantification of the carbon cycle of urban nature is needed in planning of green areas, carbon neutrality assessments, and urban climate studies. In this study, an urban land surface model SUEWS and soil carbon model Yasso were evaluated and used to estimate the carbon sequestration of street trees and soil in Helsinki, Finland. The compensation point when street tree plantings turn from annual source to sink was achieved after 14 years of the planting of the street trees. The annual carbon

sequestration depended on environmental factors such as air temperature and humidity indicating the need for modelling techniques allowing to take appropriately the local climate conditions into account. Yasso and SUEWS are able to simulate the carbon cycle of street tree plantings as shown against observed soil moisture, sap flow and soil carbon from two street tree sites, but the used substrates vary widely and the indeterminable soil properties cause great uncertainty in estimations of the longevity of soil organic carbon. However, Yasso developed for a non-urban area performs reasonably but further studies especially on

root litter input and on the role of soil moisture in the decomposition process would decrease the model uncertainties.

*Code and data availability.* The data sets are openly available at Havu et al. (2021), including the model runs for SUEWS and Yasso, the fittings of the environmental response functions, the gap filling of the meteorological measurements, and codes to reproduce the figures.

## Appendix A:  Gap filling the meteorological data

Data from two locations were used to generate the continuous meteorological data set for 2002–2016 used to force SUEWS

and Yasso models. The measurements from SMEAR III station tower and nearby roof (Järvi et al., 2009) were primary used and gap filled with measurements from Helsinki-Vantaa airport hosted by Finnish Meteorological Institute (FMI) located 10 km from Viikki. Additional SYNOP weather station precipitation measurements from Kumpula hosted by FMI were also used.

Precipitation was gap filled with multiple measurement devices and locations. The order of measurements used in the gap filling was: hourly PWD (since 2014), hourly SYNOP from Kumpula (since 2006), hourly Ott (since summer 2002), daily

SYNOP from Kumpula (since 2006) and daily SYNOP from the airport (since 2002). Daily SYNOP data were divided evenly over the day to get the hourly values.

Temperature, wind speed, wind direction and incoming radiation were measured at tower, rooftop and airport, whereas, relative humidity and air pressure only from rooftop and airport. Primary measurements were either the tower and rooftop measurements which were gap filled using airport measurements using a linear correlation. Rest of the missing hours were gap
filled by linear interpolation if less than 5 hours were missing (2 hours for radiation), or with the average of the same hour from the previous day and the following day, if less than day was missing. If more than day was missing, the values were filled by calculating the average for the same hour of three previous days and three following days.

*Author contributions.* MH, LJ and LK conceptualized the study. MH did the SUEWS and Yasso model runs, formal analysis and prepared figures. AR and PK collected the data, PK did the SPP model runs. LJ, LK and TV supervised the study. All the authors contributed to the
writing and preparation of the manuscript.

*Competing interests.* The authors declare that they have no conflict of interest.

*Acknowledgements.* The work was supported by the Tiina and Antti Herlin Foundation, Academy of Finland funded CarboCity project (decisions: 321527 and 325549) and the Atmosphere and Climate Competence Center (ACCC, decisions: 337549 and 337552), Strategic Research Council funded CO-CARBON project (decisions: 335201 and 335204), and the Tyumen region government in accordance with the
Program of the World-Class West Siberian Interregional Scientific and Educational Center (National Project "Nauka"). We also thank Toni Viskari for the introduction of the Yasso model.





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
