# Peer review of "Carbon sequestration potential of street tree plantings in Helsinki"

_Biogeosciences, 2021_

## Author Response (AR1)

**Response to the 1ˢᵗ referee**

We would like to thank the referee for his/her time, the constructive comments and suggestions. Those allowed us to greatly improve the manuscript. Please find below the detailed responses. The reviewer comments are in italics whereas our response is plain text.

*In "Carbon sequestration potential of street tree plantings in Helsinki" Havu and collaborators present model simulations over an urban area in Helsinki to estimate the carbon sequestration of urban trees. The authors used the SUEWS and Yasso models to simulate the carbon cycle for the vegetation and soil, respectively. Based on the model results, they estimated that the simulated urban area will become carbon neutral in 12 or 14 years. In my opinion, the manuscript is a valuable addition to the literature since many of the land surface model studies omit urban areas despite their growing importance. Also, the manuscript describes the most pressing points to improve modeling of energy and material cycles in urban areas such as soil carbon process in the urban area and groundwater access. However, I think the manuscript would benefit from revisions to make the method and results clearer.*

We thank the referee for highlighting the relevance of the manuscript and we agree with the suggestions to make the methods and results clearer. Below, we discuss how this will be done in detail.

*Major comments*

*The metrics used for model-data evaluation were not well justified. Four metrics were used, i.e., nRMSE, RMSE, nMBE, MBE but it is not clear what is the benefit of this multitude of metrics (what does each metric teach the reader and why are exactly those metrics important to evaluate the models at hand?). This multitude of metrics also blurs the results sections. For example, in L358, nRMSE was better at the Alnus site but MBE was poorer at the Alnus site, but both were mentioned without further explanation. What does this tell the reader about the model (deficiencies)? Also in L372, it is said that the Yasso model showed the best performance in soil3 based on RMSE but it may not be based on nRMSE. As those were not dealt with in the Discussion either, it was not clear how the use of four different metrics enriched the model evaluation.*

We agree with the reviewer that the metrics used in the manuscript need further explanation. We used two metrics RMSE and MBE, and their normalized versions. We chose RMSE to evaluate the model accuracy, as it measures the average magnitude of the error. We chose MBE to describe if there is systematic error in the possible over- or underestimate. Both RMSE and MBE are useful to evaluate model performance, although they depend on the scale. As the scales notably varied in the different datasets used in this manuscript, we chose to focus on the normalized versions as then the comparisons between the sites and years are more clear.

Improved description of the metrics will be added to the manuscript in section 2.6 as follows:
Modified text (L335):
"The normalized metrics are mainly used in the analysis as they allow comparison between different scales. The nRMSE is used to evaluate the accuracy of the models and the nMBE indicates whether the models have a systematic over- or underestimation. "

We think it is valuable for a reader to see all these four metrics in Tables 4 and 5, but in the text we will focus on normalized metrics which will make the analysis more consistent. The results sections 3.1 and 3.2 will be clarified in the revised manuscript so that reader will not get lost in all these metrics.

For transpiration (starting L358), we originally used RMSE and MBE which were better at Tilia site, because transpiration was four times smaller, than at Alnus site. Now, because the scales are different, we chose to look at normalized values in which case Alnus site performed in both cases better. This will be corrected in the revised manuscript.

For Yasso model (starting L372), RMSE was better at Alnus site, and nRMSE at Tilia site. We wanted to use the same metrics for both SUEWS and Yasso, but it is worth of noticing that the number of observations might be too small to use nRMSE for Yasso. We will clarify this in the revised manuscript. However, nMBE states the systematic under- or overestimation and thus will be more useful in this case.

Modified text (L372):

"Yasso model performance was evaluated using only four measurement points in time and therefore, the following statistical values should be treated with caution. The model performance was best in soil 3 as nMBE was lowest at both sites (Table 5). "

*The study showed that the street trees in Helsinki could become carbon neutral or slightly carbon sink 14 years after planting but it is not clear what happens after those 14 years. How long does an average street tree live?  What is the soil volume affected by a street tree? What happens with the wood of street trees once they are replaced? How much carbon is emitted in the management of street trees? How does all of this affect the life cycle carbon cycle of street trees? Thinking about the carbon balance changes after those 14 years is likely to give important insights and implications for the street tree management. One of the strengths of the model would be that a theoretical experiment is possible even though it needs several assumptions. Extending the model application would make the manuscript more suitable for Biogeosciences as it would shift the focus of the manuscript from model development to new insights in the biogeochemical cycle of street trees.*

We thank the referee for the important insights on the carbon cycle of street trees. For the revised manuscript, a simplified estimation of carbon sequestration potential throughout the street tree lifespan was made with both models. Even though old street trees do exist, the expected lifespan of a street tree is approximately only 20-30 years due to various construction works (Roman and Scatena, 2011). Therefore, the estimation was made for 30 years after the street tree transplanting. For SUEWS, both photosynthesis and plant respiration were averaged from pruned years (2008-2016) and assumed that the leaf mass and calculated averaged $CO_2$ exchange rates will remain the same for  rest of the 30-year period. Similarly, a new simulation was made to estimate the soil carbon stock for 30 year period with Yasso using averaged monthly meteorology that was calculated for the same years (2008-2016), and assuming stable root litter input. In addition, we included litter input from leaves and pruned wooden parts. As these are removed from the area of interest, the decomposition is taking place somewhere else but is naturally linked to the carbon sequestration of these tree of interest. In the revised MS, we will estimate the carbon sequestration potential of street trees as in Fig. 9, for 30 years from tree planting with the unit kg C per year per tree. Therefore, the canopy area and soil affected (25 $m^2$) will be taken into consideration.

The text in the methods section will be modified as follows:

"A simplified estimation of carbon sequestration potential throughout the expected street tree lifespan was made using both models. The expected lifespan of a street tree is approximately 20–30 years (Roman and Scatena, 2011), and therefore, the estimation was made for 30 years (2002-2032) after the street tree planting. For SUEWS, both annual photosynthesis and plant respiration were averaged from pruned years (2008–2016) and assumed  that the calculated  average rate of photosynthesis and plant respiration will continue for 2017-2032. For Yasso, an additional model run using

the average monthly mean air temperature and precipitation from the same years with stable root biomass was conducted. "

We acknowledge that this manuscript does not fully cover the whole street tree carbon cycle throughout its lifecycle as the SUEWS model is developed to describe the net $CO_2$ flux on a local scale. However, we will add more insights on the full lifecycle to Discussion as previous estimates on the carbon content of the leaves from these street trees exists. Between 2002 and 2011, the leaves would contribute cumulatively 12.5 kg C per tree of which approximately 7.3 kg C per tree still remained in 2011. The total C in pruning was 0.7 kg C per tree of which 0.6 kg C still remained in 2011 as the trees were pruned only after 2008 (Riikonen et al., 2017). We will add the new analysis to the results section in the revised manuscript and will revise the Discussion to include more insights on the carbon cycle of street trees during their lifespan.

***Minor comments***

*L43 Explain why those methods are not suitable for climate change.*

We will add a clarification on how i-Tree is not yet suitable to examine the effect of climate change on carbon cycling in Finland, as its climate conditions are adjusted for US and currently, it does not take into account the future climate.
Modified text (L43):
"However, these methods are incapable of catching the correct response of urban biogenic carbon cycle to local environmental conditions and changes in local climate, as climate conditions have been adjusted for US, and thus lack high temporal resolution. In addition, the model cannot simulate carbon cycling in future climates."

*L51 I could not find the necessity of the sentence following with 'Furthermore'. It may mean that simulating the right temperature is very important due to the interaction with urban structures. Does it?*

As the focus is on the urban biogenic carbon cycle, we explained how both photosynthesis and respiration will be modelled. "Furthermore" was just to add how respiration would be modelled, but we will remove it.
Modified text (L51):
"Plant and soil respiration is modelled to exponentially depend on air temperature. "

*L63 How come the SOC will be increased in urban soils compared to the natural environment? Don't urban structures inhibit such an process?*

Pataki et al. (2006) explains that the increase in the soil is observed in the most highly managed soils due to how much more carbon input these practises will leave to the soil. The impact is visible on parks, but in general, the structure of cities affects the soil beneath buildings and paved areas, preventing such processes. Clarification on this will be added to the MS.

*L69 Starting with 'in addition' mentioning that the information was not referred to beforehand, was confusing.*

The sentence will be modified.

Modified text (L69):
"Soil carbon decomposition depends on the size of the SOC pool, and on temperature and precipitation (Davidson and Janssens, 2006). "

*L101 It was difficult to understand how the three different soils were laid in the experimental sites.*

Clarification on this will be added to the MS:
Modified text (L101-102):
"The different soils were installed as planting pockets separated by compacted gravel at Alnus site or as continuous strip at Tilia site."

*L124, 128, 134 This content might be more suitable for the result section*

We agree. As a result,
1) sentence in L124 "Comparing the water use of the different tree species…" will be removed from methods. It is already clear from the results.

2) Sentences in L128-131 "Overall, Tilia site had higher SWC..." will be removed from the methods and added to the results section 3.1.1:
   Modified text L340-342:
   "In general, Tilia site was moister than Alnus site as also seen in the observed groundwater level. Furthermore, the catchment area at the Tilia site is large, whereas Alnus site is fed mainly with local rainfall (Riikonen et al., 2011). During the summers from 2008 to 2011, the SWC was on average 27 and 13 % for Tilia and Alnus sites, respectively."

3) Sentence in L134 "The soil carbon stock estimates…" and the following sentence "The proportion of carbon..." will be removed and the section 2.6 in the methods will be modified:
   Modified text L314-315:
   "The proportion of carbon in the LOI was assumed as 0.56 (Hoogsteen et al., 2015). "

*L143 Why were the additional measurements primarily used? Was it closer to the sites?*

The quality of the PWD measurements is much better than the old precipitation measurements with Ott. We include the reasoning in the revised MS in Section 2.3: "…and these were primarily used when available due to their better quality than the Ott measurements.".

*L155 Do you mean the air pressure?*

Yes, it will be modified to the revised manuscript.

*L258, 347 How was the value for the input (0.06) decided?*

We will add clarification to the MS that the value was chosen by sensitivity testing so that the soil doesn't limit the modelled transpiration.
Modified text L258:

"The limit was chosen by sensitivity testing such that the soil does not dry and limit the modelled transpiration. "

*L349 a slight morning maximum: unclear expression.*

We will remove the expression and modify the sentence.
Modified text L349:
"The diurnal maximum of observed transpiration reached 0.27 mm h-1 in the morning at Tilia site. "

*L356 Why only two years over four years of evaluation period were referred?*

Only two years were referred because the worst and best cases were always years 2010 and 2011. Years 2008 and 2009 always fell between these two. The years in brackets will be removed in the revised manuscript to avoid confusion.

*L365 You mean similarly, respiration is higher in Alnus?*

Yes, we will add clarification.
Modified text L365:
"Similarly, maximum respiration was higher at Alnus site (5.1 mumol m-2 s-1) than at Tilia site (3.7 mumol m-2 s-1). "

*L538 The values have a different unit from Fig. 7, which reduces connectivity.*

The soil depth was 1 m, so both kg C m-2 and kg C m-3 mean the same thing. However, we agree with the reviewer that the units should be consistent throughout the manuscript and we will change all of them to m-2, which is commonly used for fluxes. This will affect L372 and y-axis label of Figure 7.

*Table 1*
*It seems to be shown way too early as table 1 is only referred to 4 pages later.*

Table 1 is first time referred in previous page as it also describes the site and connects to Figure 1. Thus, we prefer keeping Table 1 at the original place.

*Table 4*
*Why are RMSE and MBE for SWC only for the Tilia site empty? If it is because the results*
*are normalized then the results for the Alnus site should be empty too.*

Yes, thank you for noticing this mistake. It will be modified to the revised manuscript.

*Figure 2 Function g() has not been mentioned or explained in the text or the caption.*

Thank you for noticing this mistake. In previous articles, these same functions have been named either as g or f functions, and now we had accidentally used them both here. All the f() functions will be changed to g() to match the Figure 2 in the revised manuscript.

*Figure 8*
*Not clear if 'estimated' points out simulation or observation since the term 'estimates' has been used for observations.*

We will change "Estimated" to "Simulated" to clarify that these values are modelled.

**Response to the 2nd referee**

We would like to thank the referee for his/her insightful comments and suggestions on our submitted manuscript, as they allowed us to improve it. Please find below the detailed responses. The reviewer comments are in italics whereas our response is plain text.

*This paper reports on a modeling study of two urban street tree sites in Helsinki, Finland. It exploits a multi-year time series of field observations of tree sap flow, physiology, and soils for the two sites. Such measurements are rare in urban sites and using them to parameterize the models is a great strength of this project. Previous studies have measured and/or modeled urban tree net CO2 exchange over one year to a few year's time. A significant new contribution of this study is that it couples an urban land-surface model (SUEWS), which is capable of representing photosynthetic CO2 uptake and respiratory CO2 release by the tree canopy in response to environmental drivers, with a soil carbon model (YASSO) which is capable to soil organic carbon fluxes and pools for the same conditions.*

*The modeling system and is parameterization are well documented and appropriately validated. The paper is well organized and generally well written (but see note below). The manuscript and its conclusions could be strengthened by moderate revisions, which are noted below.*

We thank the referee for highlighting the novelty of the manuscript. Below, we discuss his/her suggestions to strengthen the manuscript in detail.

*General comments:*
*1. Please explain more about how the study being on juvenile trees affects your overall conclusions. Plant relative growth rate will change as the tree size gets larger. What is the typical longevity of these tree species-- in "nature" and also what is typical maximum age in the urban environment? Given the way the paper is framed around urban tree C sequestration potential and management for climate, it is likely that some readers could misunderstand or incorrectly extrapolate the findings to a mature urban forest or to the lifetime of the trees. It would help a lot if these points could be discussed when you are interpreting the main messages of your conclusions, including adding cautions or caveats where appropriate.*

We agree with the referees suggestion to add more cautions, as the studied street trees do not represent the whole variety of urban trees. In the revised MS, we will clarify that the expected lifespan of a street tree is approximately only 20-30 years, and we will also add a simplified estimation of carbon sequestration potential throughout this expected street tree lifespan by both models (see also the response to the comments by referee #1). Methods section will be modified to include description of the simplified estimation:
"A simplified estimation of carbon sequestration potential throughout the expected street tree lifespan was made using both models. The expected lifespan of a street tree is approximately 20–30 years (Roman and Scatena, 2011), and therefore, the estimation was made for 30 years (2002-2032) after the street tree planting. For SUEWS, both annual photosynthesis and plant respiration were averaged from pruned years (2008–2016) and assumed that the calculated average rate of photosynthesis and plant respiration will continue for 2017-2032. For Yasso, an additional model run using the average monthly mean air temperature and precipitation from the same years with stable root biomass was conducted."

Analysis concerning the new estimations will be added to the results section (L404). We will also revise the Discussion to include more insights on the carbon cycle of the street trees during their lifespan.

*2. The aim of the modeling integrations to estimate the tree/soil C budget is clear, and it is a novel contribution. In fact, it is really something like site-based net C exchange, and perhaps it is not wrong to call it "sequestration potential". However, again for a general audience and for land managers who would read your work, it would be very helpful to explain how this relates to carbon sequestration as a climate concept. There, we normally think of carbon sequestered as being removed from the atmosphere and stored for a climatically relevant length of time (w.r.t. to fossil C emission reductions), such as 50-100 years or longer. How does a climatic concept of C sequestration relate to the urban system that you have modeled here? How long would the trees live or be allowed to grow on site before they grow too large (height interfering with wires, roots interfering with pavement, etc.), or before they die from insect outbreaks, urban heat, drought, road salting in winter, mechanical damage, etc.? What is the normal replacement interval for street trees like this? I am not asking that values be added for all of these factors; however, it would strengthen the paper if you can explain more specifically in what ways your results could relate to long-term carbon sequestration, and in which ways they do not.*

As mentioned in our previous comment, we will add a simplified estimation on how much carbon these street trees can potentially sequester in their expected lifespan (30 years). Even though old street trees exist, road and other constructions, and renovations take place usually before street trees reach maximum size and age. We acknowledge that this manuscript does not cover all possible species and life spans, nor the full life cycle of the trees including the time in nursery, transport and construction nor the end use of the wood, as the SUEWS model is only able to produce the net $CO_2$ flux on local scale and Yasso the local soil carbon storage. Full life cycle assessment is thus out of the scope of the current manuscript. However, we thank the reviewer for the idea, and we will add more insights on this to the Discussion. The discussion will answer in what way the results relate to long-term carbon sequestration, and in which ways they do not, and what kind of restrictions there is in upscaling these results. In the end, the aim of the paper is to introduce and validate the models, and use the models to examine carbon cycling of two street tree species. Thus, an application on a larger spatial scale is out of the scope of this paper.

*3. It's understood that this is a modeling study, but it would strengthen the paper if the discussion included some comparison of your results to other field or modeling based studies of urban annual net biogenic (tree/soil) C exchange. There are some from northern climates such as Vancouver, Minneapolis, London?, even Helsinki. Broadly, how do the conclusions here compare to those obtained for tree-covered landscapes in cities that have been obtained through flux measurements and/or model upscaling?*

We acknowledge the referees suggestion to include more net biogenic C exchange studies to the manuscript. However, we would like to have the focus only on street trees, for which such studies to our knowledge do not exist. Therefore, we have compared the SUEWS model results with other street tree models in Europe, and the soil carbon model with respiration and soil carbon stock measurements (especially in section 4.1). In the revised MS, we will highlight that there are not comparable studies available.

*4. Please check for consistence of verb tense throughout the paper. In places it switches back and forth between past and present tense.*

We thank you for this suggestion. We will revisit the manuscript throughout. Particularly corrections will be made in sections *2.4.1, 2.4.2, 2.4.3*.

***Detailed comments:***

*Does the soil freeze to a significant depth in winter at these sites? How was frozen soil handled in the YASSO simulations (does Rsoil decline or even stop)?*

Usually, the sites have snow cover that protects the soil from freezing in winter and it is quite common that even if there is some ice formation, notable share of the soil water is in liquid phase and the soil temperature stays close to zero. In such conditions, the decomposition of soil organic matter is slow but still existing. However, total freezing of soil is possible in low air temperature if there is no snow cover but there is no mechanism in Yasso that would consider it. Instead, the decomposition rate follows just the changes in air temperature also in frozen conditions. In any case, such episodes when soil is deeply frozen are rare in Helsinki and as the time step is only one month, such episodes are insignificant. We will describe the model behaviour in freezing circumstances in the revised manuscript.

Modified text in section 2.5 (starting L274):

"Air temperature goes below freezing during the studied period but typically the snow cover prevents soil from freezing. Even if there would be some ice formation in the soil, notable share of the soil water would still be in liquid phase and the soil temperature stays close to zero. Also, in Yasso there is no mechanism to account for completely frozen soil. Thus, in the model runs the decomposition rate follows the changes in air temperature also in frozen conditions."

*line 129: Do you know how high was the groundwater table at the two sites? Did the level vary by season and, if so, how would that have affected the results? And a related question: Was any irrigation used at the sites? (I am assuming not regularly because it was not mentioned in the manuscript.) However, was it used during the early years of the trees' growth--it is common for irrigation to be needed in the first 2-3 years after establishment, depending on the local precipitation regime.*

At Tilia site, previously undetected high groundwater table was found during the street construction, and the level varied between 50-90 cm below street level. However, we did not have any information of the seasonality, therefore, we don't know how this would have affected the observations. The street trees were irrigated weekly for two years after transplanting, but we did not take this into account in our model runs. SUEWS has an irrigation model that can be turned on/off but it is designed for garden irrigation and not for this kind of weekly irrigation. Current version of Yasso cannot take irrigation into account at all. We will add a clarification to the study site description about the irrigation.

Modified text (L98):

"The trees were irrigated weekly for two years after street construction. Irrigation was however neglected in the model simulations as Yasso cannot currently include irrigation and the irrigation model in SUEWS is designed for typical garden irrigation. This is expected to have a minor impact on the results. "

*line 282-3: Aboveground litter was ignored in the simulations on the basis that the autumn leaf fall of these deciduous species is normally removed. This is a reasonable approach for modeling the C exchange of the "tree site" itself. However, I think you should take this issue further in the discussion and conclusions of the paper because you have "framed" the paper around urban tree plantings and sequestration. What would be the consequence of leaf litter fall for your annual carbon budget and how would this affect your overall conclusions and implications for how urban tree plantings affect the urban C budget?*

We will add more insights to Discussion on the carbon content of the leaves as there are previous articles from these street trees that have already estimated some values. Between 2002 and 2011, the leaves would contribute cumulatively 12.5 kg C per tree of which approximately 7.3 kg C per tree still remained in 2011. The total C in pruning was 0.7 kg C per tree of which 0.6 kg C still remained in 2011 as the trees were pruned only after 2008 (Riikonen et al., 2017). As mentioned previously, we will add a simplified estimation of carbon sequestration potential throughout expected street tree lifespan and include the soil respiration estimate arising from the decomposition of leaves and pruned wooden parts.

*line 321: Canopy densification was ignored (stopped) in the model after a certain year, based on the fact that these trees were pruned annually after they had reached that age. However, the biomass of leaves and branches removed by pruning would presumably be used to create mulch or compost or biofuel, etc, thereby all being released to the atmosphere. So, similarly as in the comment immediately above, how would the exclusion of the pruned biomass affect your annual carbon budget and what are the implications for your overall conclusions? If it is possible to make a quick quantitative estimate of these two carbon losses (collected litter and pruned biomass), that would be a nice addition that would strengthen the paper. If it is not possible to have a quantitative value, then it would at least be good to add these points to the discussion and the explanation of conclusions about the total C budget from urban trees.*

As mentioned previously, we will add more discussion on this to the revised manuscript, although the effect of pruning seemed small for these young trees (cumulatively 0.7 kg C per tree in 2003-2011) compared to the carbon in leaves and even our estimates. However, the trees were pruned only after 2008, which would affect the future estimations as the trees grow. To support the discussion, we will calculate soil respiration estimate arising from the decomposition of leaves and pruned wooden parts. As these are removed from the area of interest, the decomposition is taking place somewhere else but is naturally linked to the carbon sequestration of these tree of interest.

*line 398: "climate neutral" is not quite correct, in my view. First, there are, of course, other effects of trees on climate besides net CO2 exchange. Second, there is the point about the model simulations being focused on the tree/soil system. I'd suggest writing "carbon neutral", and with the caveat that it's from the point of view of the tree/soil system (without the exports of leaf litter and pruned biomass).*

We agree with the reviewer, that carbon neutral is more suitable term, and this will be changed to the revised manuscript.

*line 426: When you discuss year-to-year variability, can you also say something about how important was the effect of annual differences in growing-season length, or timing? Especially in cold climates, these two factors can be important for annual C exchange, beyond only the variation in Tair.*

We thank the reviewer for pointing this out. In the result section we noted on L386 that "Leaf onset begun at different times in different years depending on the simulated growing degree days, leading to a difference of up to 20 days in the model simulations." We extended our analysis for the whole growing season, and based on simulated LAI, the active season varied up to 26 days in the model simulations. We will add these details in the revised MS and extend the discussion also in section 4.1 (L426). However, as the weather conditions vary during the growing season despite its length so

much it is not possible to derive conclusions on the exact impact of the growing season length on carbon sequestration. However, weather conditions vary throughout the season so with these model runs utilizing the measured weather, it is not easy to determine the effect of growing season length to the photosynthetic production of the whole year. We agree that it is an interesting point and we are happy to discover that in further studies where we or some other group run the model for other applications but here, we would like to keep current focus on model testing and simulating the initial development of a street plantation.

---

## Author Response (AR2)

**Response to editor comments**

We would like to thank the editor for his time and constructive comments. We took another look at the manuscript and improved it according to the suggestions. Please find below the detailed responses.

*Dear,*
*Thank you for submitting a revised version of your manuscript. Although the revisions address the main concerns of the referees, I would like to encourage you to have another look at the manuscript to further improve the following issues:*

*(1) The urban environment is extremely heterogenous and this is also true for its soils. In the method sections the soil conditions are detailed but this information is not referred to in the abstract or the discussion. Expressing the sink and sources at the tree level is an elegant way of dealing with this issue. Both the problem and solution could be better stressed in the discussion.*

As suggested, we added into the abstract that we studied the trees in the three different growing media. In addition, we modified the following to abstract (L15):

> "The models were able to capture the variability in urban carbon cycle and transpiration due to changes in environmental conditions, soil type, and tree species. Carbon sequestration potential was estimated for an average street tree and for the average of diverse soils present in the study area."

In addition, we highlighted the problem of diversity and the solution used in this study in the beginning of the second paragraph in the discussion (L456):

> "Urban areas are heterogeneous with variation in soil properties, plant species, and biomass. Even streets have diverse soil types, making it difficult to assess the carbon sequestration potential of street tree plantings. Here, we estimated the sequestration potential for street trees by utilizing an average calculated over diverse soil types and taking into account the most common city-wide planting pocket size for street trees ($25 \ m^2$). "

*(2) The heterogeneity of the urban environment is a bit downplayed in the introduction. It is mentioned but the terminology is vague, i.e., L29-30. Add the context of these studies (temperate/tropical/boreal cities with X inhabitants and Y% green spaces). The cited 14% is context-dependent. The next sentence makes this clear but it is not clear when/where this 14% has been observed.*

We agree and we clarified the context of the studies on L29 in the revised MS:

> "Urban green areas have been found to sequester significant levels of city GHG emissions. For example, the biogenic carbon fluxes in Boston, USA, and Florence, Italy amounted to 14% (Hardiman et al., 2017) and 6.2% (Vaccari et al., 2013) of both cities' GHG emissions, respectively."

After these sentences, we stressed the diversity by mentioning variation both in soil and in plant species and biomass.

*(3) The abstract still contains three different error-measures. This is confusing and has a flavour of cherry picking. The error-measures are explained after the abstract. Are the three error values so important that they should be mentioned in the abstract? Note that Biogeosciences aims to publish biogeochemical studies. For Biogeosciences, the model application is more interesting and should*

*be the focus of the abstract. The model development and evaluation is the tool but not the most interesting result. This could be better reflected in the abstract and discussion.*

We agree that the error-measures are not any of our key messages and those do not need to be mentioned in the abstract. Thus, the following sentences were removed as they truly are more model development and evaluation and not application (L15-17).

> "SUEWS simulated the stomatal control and transpiration well (RMSE<0.31 mm h$^{-1}$) and was able to produce correct soil moisture in the street soil (nRMSE<0.23). Yasso was able to simulate the strong decline in initial carbon content but later overestimated respiration and thus underestimated carbon stock slightly (MBE>-5.42 kg C m$^{-2}$)."

*(4) Which processes should be included in urban models that are currently not accounted for in the forest models? If it is just different climate, the current forest models are capable of dealing with that. The text hints at the interactions between the vegetation and the built-up area but it is not clear which interactions are accounted for. It remains also unclear how the Yasso model (which was developed in a forest context) was adjusted for the urban environment. Consider adding a table listing the underlying assumptions and specific urban-processes that were accounted for in this study. Such a table could help the readers to better understand how your approach differs from simply running a forest model with an urban climate forcing.*

As suggested, we clarified the urban processes by adding a sentence to clarify the interaction between built and vegetative surfaces to introduction, L49:

> "In reality, the built environment in urban areas allows the formation of the urban heat island effect, strong variation in soil moisture, and lateral water flows between built-up and vegetative surfaces. "

In addition, we added a new Appendix B: "Specific urban processes used in $CO_2$ models", which includes a table that clarifies the specific urban-processes that were accounted in this study.

*(5) As a non-English speaker myself, I do appreciate the difficulty in expressing ones thinking in another language. The English in the manuscript is readable but there are still several grammar issues and awkward sentences. A native speaker could easily overcome these issues. Correct language is likely to broaden the readership and to increase the chances that the paper gets cited.*

The manuscript has now gone through the language revision by a native speaker working in University of Helsinki language services.

*Specific issues*
*(6) There is no page limitation so avoid these nontrivial acronyms LOI, BD, LA, NE, ... they are a burden for the reader*

We have tried now minimize the use of acronyms:
1. BD and LA are removed as they are only mentioned once.
2. LOI is now used if it appears multiple times in one section. Otherwise, the acronym is not used.
3. NE is now used only for Figure 9.

*(7) Table 1. All acronyms should be explained in the caption.*

The variable names are modified so that there are not acronyms.

*(8) L87. This is methods and should be moved to the methods.*

The sentence was removed. It is already clear from the methods.